# AlphaZero in Sparsely Rewarded Games: Limits and Auxiliary Supervision

## Abstract

AlphaZero has demonstrated that a neural-guided Monte Carlo Tree Search can achieve superhuman performance, but strong play does not necessarily imply perfect play. We study this gap in two oracle-evaluable domains with contrasting structure: Connect Four, a solved partisan game with exact game-theoretic values, and Chomp, an impartial game whose optimal play is governed by Grundy-number structure. Under a unified self-play + MCTS pipeline, we compare vanilla AlphaZero, a multi-frame variant (limited to Chomp), and an AlphaZero Auxiliary Loss (AZAL) that adds oracle-derived policy supervision. We find that vanilla AlphaZero achieves strong play across both domains but cannot preserve the exact trajectories required for optimal play: in Connect Four, it fails to maintain the optimal line of play, while in Chomp, it fails to consistently restore the $g = 0$ invariant. On rectangular Chomp boards, multi-frame inputs alone do not remove this gap. Nevertheless, AZAL substantially improves oracle consistency across multi-seeded full-game traces and sampled-state evaluations. On Chomp, AZAL reaches perfect full-game oracle consistency on $10 \times 11$ and high but not complete consistency on $9 \times 10$; on Connect Four, AZAL improves oracle-match rate and delays the first oracle mistake, but does not reach perfect play. Our codes can be found at: `https://anonymous.4open.science/r/AlphaZero-Auxiliary-Loss-F74E/readme.md`

## 1 Introduction

AlphaZero established the Monte Carlo Tree Search (MCTS) as a central paradigm in modern game AI. Building on the broader ideas of planning and generalization, AlphaGo Zero showed that search can be embedded directly into the reinforcement learning loop, improving policy targets while self-play supplies value supervision (Anthony et al., 2017; Silver et al., 2017). AlphaZero then generalized this recipe across Go, chess, and shogi, illustrating that a single policy–value network guided by MCTS can achieve superhuman play from random initialization, provided only the rules of the game (Silver et al., 2018).

Yet superhuman play is not identical to perfect play. An AlphaZero-style system may achieve excellent results from the standard starting position while fail to choose the optimal move. This distinction has become especially important in games that depend on sparse global features and not dense local tactics. In such settings, errors in policy and value estimation do not remain isolated: they affect the search targets used to generate future self-play data. This weakens the positive feedback loop on which AlphaZero depends (Zhou & Riis, 2022; Riis, 2024).

This paper analyzes the gap between strong play and perfect play through two contrasting domains: Connect Four and Chomp. Connect Four is a solved partisan game with exact game-theoretic values, allowing direct comparison to winning trajectories (Allis, 1988). By contrast, Chomp is an impartial combinatorial game whose winning structure is naturally studied through Grundy numbers (Sprague, 1935; Grundy, 1939). Despite classical theoretical results, the explicit optimal strategies of Chomp remain difficult to characterize (Gale, 1974; Brouwer et al., 2005). Together, these games provide a useful test of whether AlphaZero fails to recover exact play across games with very different strategic structure.

We provide an empirical study of AlphaZero-style learning on both domains under a standard self-play + MCTS pipeline. We evaluate vanilla AlphaZero against exact oracles, compare it with a multi-frame variant

inspired by Riis (2024) (limited to Chomp), and introduce an AlphaZero Auxiliary Loss (AZAL) variant designed to provide a stronger learning signal. Across both domains, multi-seed and sampled-state oracle evaluations reveal that vanilla AlphaZero remains far from exact oracle consistency. Multi-frame inputs do not resolve the Chomp rectangular-board failure in our experiments. AZAL substantially improves oracle consistency, achieving perfect full-game consistency on Chomp 10×11, high but incomplete consistency on Chomp 9×10, and decent but still imperfect improvement on Connect Four.

The broader lesson is that search-improved self-play can produce policies that are empirically strong yet structurally misaligned with exact optimality, and oracle-evaluable games provide a useful diagnostic setting for separating these two notions.

Our contributions are as follows:

1. We show that vanilla AlphaZero can learn strong self-play policies without routinely recovering oracle-consistent play in Connect Four and Chomp, as measured by exact oracle evaluation over multi-seed full-game traces and randomly sampled states.

2. We compare two possible remedies under a unified self-play + MCTS framework: a multi-frame representation (for Chomp) and an auxiliary-loss variant, AZAL.

3. We find that AZAL substantially improves oracle consistency across full-game and sampled-state evaluations, especially in Chomp, suggesting the standard AlphaZero search-learning signal is a plausible bottleneck for recovering exact play.

## 2 Related Work

### 2.1 AlphaZero and expert iteration

AlphaZero belongs to an extensive line of work that combines planning with function approximation in self-play reinforcement learning. Expert Iteration utilizes a tree search as a strong expert and a neural network as a fast learner (Anthony et al., 2017). AlphaGo Zero integrated this idea into an end-to-end self-play framework, wherein MCTS improves policy targets while game outcomes supervise value learning (Silver et al., 2017). AlphaZero then generalized the same recipe across Go, chess, and shogi, demonstrating that a single search-guided policy–value network can achieve superhuman performance without handcrafted evaluation or human demonstration data (Silver et al., 2018).

### 2.2 Strong play versus perfect play

Strong empirical play does not imply convergence to perfect play. Zhou and Riis argue that AlphaZero-style agents may become effective *champions* while fail to become true *experts*, especially in impartial games whose optimal behavior depends on sparse global structure that is difficult to infer from self-play alone (Zhou & Riis, 2022). Riis further suggests that single-frame representations are insufficient for recovering such structure in games like Nim and proposes multi-frame inputs as one possible remedy (Riis, 2024). Relatedly, Trudeau and Bowling identify search inefficiencies in standard AlphaZero-style training, where self-play beginning from the initial position can weaken supervision for deeper vital states. Go-Exploit targets deep-state undersupervision and sample efficiency (Trudeau & Bowling, 2023). However, our work differs as it diagnoses oracle consistency move-by-move instead of proposing a sample-efficient training schedule. Thus, the goal is not to beat Go-Exploit but to test if standard self-play recovers exact oracle trajectories.

Together, this literature motivates evaluating whether AlphaZero actually recovers the exact structural regularities required for perfect play.

## 3 Problem Setting and Oracle Evaluation

Our goal is to study the disparity between *superhuman* and *perfect* play in AlphaZero-style agents. We do so in two oracle-evaluable domains with contrasting structure: Connect Four and Chomp. In both games, the oracle allows us to assess if the learned agent preserves the trajectories required by optimal play.

### 3.1 Superhuman vs. perfect play

In this paper, we distinguish between *superhuman* and *perfect* play. By superhuman play, we mean empirical performance that exceeds strong practical baselines or human-level play. By perfect play, we mean selecting an optimal move from every legal state, as judged by an exact solver or oracle. This distinction is central to our study: an AlphaZero-style agent may achieve high win rates but fail to preserve the oracle-optimal trajectories required for exact play. In Connect Four, both players can be evaluated against exact game-theoretic value, whereas in Chomp, players are evaluated by their ability to preserve the winning invariant by moving to $g = 0$ states.

### 3.2 The games

Connect Four and Chomp are both perfect-information, turn-based board games, but they differ sharply in their rules and strategic structure. In Connect Four, two players alternately drop discs into the columns of a vertical $6 \times 7$ grid. The objective is to be the first to align four of one's own discs horizontally, vertically, or diagonally.

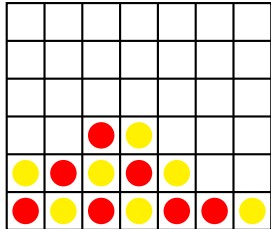

Figure 1: A sample Connect Four position with red and yellow discs on the standard $6 \times 7$ grid.

In Chomp, play begins from a rectangular grid of squares. A move selects a remaining square and removes that square together with all squares below and to its right. Consequently, the board shrinks monotonically over time. Under normal-play convention, the player forced to take the poisoned, upper-left corner square loses. These contrasting rules make the games useful testbeds for comparing AlphaZero-style learning in partisan and impartial settings.

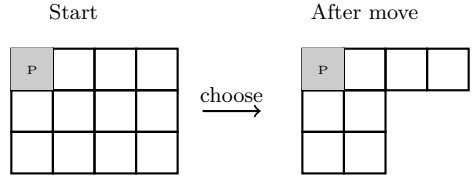

Figure 2: A small Chomp progression on a $3 \times 4$ board. Selecting a square removes that square together with all squares below and to its right; the poisoned corner $P$ must be avoided.

### 3.3 Connect Four oracle

For Connect Four, we use the perfect solver of Pascal Pons (Pons, 2015). Since Connect Four is a partisan game, the natural oracle quantity is an exact *game-theoretic score*. Let $\sigma(s, a)$ denote the exact score of taking move $a$ in state $s$, and let

$$m(s) := \text{nbMoves}(s) \tag{1}$$

denote the number of discs already played in position $s$. The solver computes

$$\sigma(s,a) = \begin{cases} \dfrac{WH+1-m(s)}{2}, & \text{if } a \text{ wins immediately,} \\ -v\big(T(s,a)\big), & \text{otherwise,} \end{cases} \tag{2}$$

where $W$ and $H$ are the board dimensions, and $T(s,a)$ is the successor position after applying move $a$ (Pons, 2015). The exact value of a state is then

$$v(s) = \max_{a \in \mathcal{A}(s)} \sigma(s,a), \tag{3}$$

where $\mathcal{A}(s)$ is the set of legal actions.

Positive values correspond to forced wins, negative values to forced losses, and zero to forced draws. The magnitude reflects distance to resolution: larger positive scores equate faster wins, while larger (less negative) losing scores equate slower losses. Additional solver and encoding details are deferred to the appendix A.1.

### 3.4 Chomp oracle

For Chomp, we use a recursive *Grundy-number solver*. Because Chomp is an impartial normal-play game, every state $s$ admits a Grundy number $g(s) \in \mathbb{N}$, with $g(s) = 0$ if and only if the position is losing for the player to move (Sprague, 1935; Grundy, 1939). The oracle is defined recursively by

$$g(s) = \text{mex}\big(\{g(s') : s' \in \mathcal{N}(s)\}\big), \tag{4}$$

where $\mathcal{N}(s)$ is the set of legal successors and $\text{mex}(\cdot)$ is the minimum excluded nonnegative integer. From any winning position ($g(s) \neq 0$), an optimal move is one that reaches a successor with Grundy number zero.

In the main text, this oracle is used to determine if the transition $g(s) \neq 0 \rightarrow g(s') = 0$ is preserved on winning turns. Board encoding, move indexing, and memoization details are deferred to the appendix A.2.

## 4 Methods

We compare three closely related AlphaZero-style agents: vanilla AlphaZero, a multi-frame variant (restricted to Chomp), and AlphaZero-Auxiliary Loss (AZAL). In all cases, self-play games are generated with MCTS guided by a policy–value network, replay tuples are collected from those games, and the network is updated from search-improved policy targets and game outcomes. The methods therefore differ only in the representation provided to the network and in the strength of the supervision signal.

**Vanilla AlphaZero.** Our baseline follows the standard AlphaZero recipe (Silver et al., 2018): a policy–value network is trained from self-play data, where MCTS produces improved policy targets and terminal outcomes provide value targets. The policy is learned only through search-improved self-play. Full architecture, search, and optimization details are deferred to Section B.1.

**Multi-frame AlphaZero.** To test if the main bottleneck derives from state representation, we evaluate a multi-frame variant inspired by Riis (2024). The training loop, search procedure, and optimization objective are identical to vanilla AlphaZero; the only difference is that the network receives a short stack of recent states instead of a single board snapshot. We therefore treat multi-frame AlphaZero as a representation ablation rather than a distinct learning algorithm. See Section B.2.

**AlphaZero-Auxiliary Loss (AZAL).** To strengthen the training signal, AZAL augments the standard AlphaZero objective with an oracle-derived auxiliary policy loss:

$$\mathcal{L} = \mathcal{L}_{\text{policy}} + \mathcal{L}_{\text{value}} + \lambda_{\text{aux}}\mathcal{L}_{\text{aux}}.$$

This auxiliary term favors oracle-consistent actions during training, but leaves self-play, MCTS, and value targets unchanged. Thus, AZAL is a minimal modification of vanilla AlphaZero that changes supervision,

not search. For labeled states, the auxiliary term is a cross-entropy loss against a softened oracle target distribution

$$q(a \mid s) \propto \big(\mathbf{1}[a \in \mathcal{B}(s)] + \varepsilon\big)\mathbf{1}[a \in \mathcal{A}(s)],$$

where $\mathcal{B}(s)$ is the oracle-optimal moveset and $\mathcal{A}(s)$ is the legal action set. We use auxiliary-target smoothing $\epsilon = 10^{-3}$ for all AZAL experiments. The per-state auxiliary loss is then

$$\mathcal{L}_{\mathrm{aux}}(s) = -\sum_a q(a \mid s) \log p_\theta(a \mid s),$$

where $p_\theta(a \mid s)$ is the policy distribution predicted by the network for state $s$, obtained by applying a softmax to the policy-head logits. This loss is averaged over labeled states only. Thus, AZAL biases the policy head toward oracle-consistent actions during training without changing the search procedure itself. Domain-specific label construction in Connect Four and Chomp is deferred to Section B.3.

**Experimental protocol.** All methods use the same self-play + MCTS loop and search budget, so comparisons distinguish the effects of representation and supervision rather than changes in compute or replay generation. Evaluation uses exact oracles: the perfect solver in Connect Four and the Grundy oracle in Chomp. In addition to aggregate losses, we analyze deterministic self-play trace rollouts and evaluate them with exact oracle annotations to distinguish strong empirical play from exact optimality. Only the final checkpoints are evaluated; checkpoint sensitivity is left for future work. Full protocol and logging details are given in Section B.4.

**Trace-level metrics.** To summarize oracle alignment compactly, we report three shared trace-level metrics: *oracle-match rate* (Match), the fraction of oracle-labeled moves that lie in the oracle-optimal action set, and the *longest oracle-consistent chain* (Chain), defined as the longest contiguous run of oracle-consistent moves within the rollout, and *first non-oracle ply* (FirstFail), defined as the first move that does not follow oracle supervision. These quantities are reported separately for the first and second player. The main aggregate trace-level metrics appear in Table 1, player-specific match rates appear in Table 4, and representative trace panels are shown in Figure 3.

We report both aggregate and representative trace evaluations. The main full-game results aggregate 60 self-play traces across three seeds per model and game setting. We also report sampled-state oracle evaluation from random-start positions to test if the observed behavior persists beyond opening rollouts. Sample states are generated from random legal moves and sampled across three seeds. They are evaluated only when an oracle-best moveset is defined, evaluating behavior beyond the standard opening rollout.

The deterministic greedy traces in Figure 3 and Tables 6-8 are retained as representative oracle-annotated examples.

Table 1: Multi-seed full-game oracle consistency across model variants. Each row aggregates 60 full-game self-play traces across three seeds. Perfect denotes the fraction of traces with no labeled oracle mistake. Match denotes pooled oracle-match rate over all labeled plies. Chain denotes the mean longest oracle-consistent chain per trace. FirstFail denotes the mean first labeled non-oracle ply, excluding perfect traces.

| Game | Model | Traces | Perfect | Match | Chain | FirstFail |
|---|---|---|---|---|---|---|
| Chomp 9×10 | Vanilla | 60 | 0.000 (0/60) | 0.609 (487/800) | 7.867 ± 1.565 | 0.000 ± 0.000 |
| | Multi-Frame | 60 | 0.000 (0/60) | 0.483 (272/563) | 4.400 ± 1.846 | 0.000 ± 0.000 |
| | AZAL | 60 | 0.567 (34/60) | 0.948 (1065/1123) | 13.450 ± 2.801 | 11.923 ± 1.035 |
| Chomp 10×11 | Vanilla | 60 | 0.000 (0/60) | 0.474 (484/1021) | 7.550 ± 1.175 | 0.000 ± 0.000 |
| | Multi-Frame | 60 | 0.000 (0/60) | 0.463 (317/685) | 5.217 ± 1.292 | 0.000 ± 0.000 |
| | AZAL | 60 | 1.000 (60/60) | 1.000 (793/793) | 13.217 ± 1.392 | N/A |
| Connect Four | Vanilla | 60 | 0.000 (0/60) | 0.785 (1700/2165) | 17.000 ± 8.325 | 0.867 ± 1.310 |
| | AZAL | 60 | 0.000 (0/60) | 0.849 (1828/2153) | 16.967 ± 5.335 | 5.283 ± 3.933 |

Table 2: Random-start sampled-state oracle evaluation. Positions denotes the number of sampled states evaluated across three seeds. Labeled denotes the number of sampled states with a defined oracle-best moveset. Match denotes pooled oracle-match rate over labeled sampled states. RunMean reports the mean and standard deviation of sampled-state match rate across evaluation runs.

| Game | Model | Positions | Labeled | Match | RunMean |
|------|-------|-----------|---------|-------|---------|
| Chomp 9×10 | Vanilla | 36 | 34 | 0.500 (17/34) | 0.481 ± 0.114 |
| | Multi-Frame | 36 | 34 | 0.176 (6/34) | 0.171 ± 0.051 |
| | AZAL | 36 | 34 | 1.000 (34/34) | 1.000 ± 0.000 |
| Chomp 10×11 | Vanilla | 39 | 35 | 0.400 (14/35) | 0.381 ± 0.135 |
| | Multi-Frame | 39 | 35 | 0.286 (10/35) | 0.267 ± 0.134 |
| | AZAL | 39 | 35 | 0.829 (29/35) | 0.816 ± 0.093 |
| Connect Four | Vanilla | 56 | 56 | 0.589 (33/56) | 0.587 ± 0.147 |
| | AZAL | 56 | 56 | 0.768 (43/56) | 0.769 ± 0.097 |

## 5  Vanilla AlphaZero: Strong Play Without Perfect Play

### 5.1  Connect Four

Vanilla AlphaZero learns a strong Connect Four policy but does not reliably recover oracle-consistent play. The main failure is an inability to *preserve* the optimal moveset. This distinction is clearest at the trace level.

**Trace evidence.**  The self-play rollout in Table 8a, together with the score trajectory in Figure 3a, demonstrate that vanilla AlphaZero does not reliably preserve the winning continuation. Visually, Figure 3a shows repeated oscillation across positive, zero, and negative score regions.

Table 1 reinforces these trends. Although the baseline model's oracle-match rate is 0.785 (1700/2165), its first labeled failure occurs very early on average (0.867 ± 1.310), indicating that the model is unable to discern the correct opening move. Table 2 confirms the gap beyond the opening trace: the vanilla model matches the oracle on only 0.589 (33/56) of sampled Connect Four positions.

The key failure is therefore not an inability to play locally strong moves. Rather, vanilla AlphaZero repeatedly leaves the optimal continuation and relinquishes its winning edge. Furthermore, the extended sequence of neutral moves in midgame (Figure 3a) suggests the model struggles to identify winning moves. This is precisely the gap between strong play and perfect play in Connect Four: the model produces tactically plausible play, but it does not preserve the small set of early value-preserving decisions required for exact optimality.

**Training curves.**  The training curves are consistent with this interpretation; see Section D. Policy loss falls rapidly during early training, then enters a flatter regime with persistent oscillations. This indicates that the network learns a substantially stronger move distribution but does not cleanly correspond to its evolving MCTS targets. The value loss exhibits a similar two-stage pattern, but with even stronger oscillatory behavior: although it drops to a relatively low absolute magnitude, it continues to spike sharply throughout training. The total loss inherits both stagnation and oscillation, suggesting the network is repeatedly chasing search targets generated by an evolving self-play distribution.

### 5.2  Chomp

#### 5.2.1  Square-board sanity check

Before turning to the harder rectangular settings, the square-board traces in Tables 6a and 6c serve as an important sanity check. On both $9 \times 9$ and $10 \times 10$, vanilla AlphaZero self-play exhibits the Grundy-optimal alternation: when a winning move exists, Player 1 moves from $g(s) \neq 0$ to $g(s') = 0$, after which Player 2 moves back to a nonzero-Grundy state. In these square-board traces, AlphaZero matches the oracle reference move on every winning turn. Thus, on these square instances, the overall pipeline—including move encoding,

oracle integration, and trace logging—is consistent with exact Grundy-optimal play. This makes the failures on rectangular boards more meaningful: the issue is that the learned policy stops preserving the invariant as the task becomes harder.

### 5.2.2 Rectangular-board failure

On the rectangular boards, the picture changes sharply across both representative traces and aggregate evaluations. Vanilla AlphaZero no longer reliably preserves the winning invariant, and the gap between strong and perfect play becomes immediate and structural rather than occasional.

**Trace evidence.** The self-play traces on $9 \times 10$ and $10 \times 11$ in Tables 6b and 6d, together with the trajectories in Figure 3b–c, show that vanilla AlphaZero does *not* preserve the $g = 0$ manifold once winning opportunities arise.

From Table 1, vanilla AlphaZero matches the oracle 60.9% of the time on $9 \times 10$ and 47.4% on $10 \times 11$. The model's first failure occurs immediately on average: FirstFail = 0.000 on both boards. Sampled-state evaluation confirms a similar pattern (Table 2): the baseline model reaches a match rate of only 0.500 on $9 \times 10$ and 0.400 on $10 \times 11$.

Figure 3b and Figure 3c make this failure visually explicit. The early plies contain non-oracle moves from winning positions, after which the rollout no longer alternates cleanly between nonzero and zero Grundy states. A characteristic pattern emerges: AlphaZero sometimes makes correct moves in the early game, but only near the late game—after the board has been reduced substantially—does play become consistently oracle-aligned.

**Training curves.** The training curves support the same diagnosis; see Figure 4, especially the policy, value, and total loss panels for the rectangular-board settings. On $9 \times 10$ and $10 \times 11$, policy loss decreases rapidly at first and then plateaus at relatively high levels, indicating that the network is learning to imitate its own MCTS-improved targets without collapsing onto a sharply concentrated oracle-like policy. The value loss is more concerning: compared with the easier $9 \times 9$ and $10 \times 10$ settings, it saturates at a distinctly higher level and remains persistently noisy rather than converging toward zero. The total loss mirrors this behavior, showing apparent early progress followed by a prolonged plateau rather than stable convergence.

## 6 Strengthening the Signal: Multi-frame and AZAL

The vanilla results suggest that weakness in the standard search-learning signal is a plausible bottleneck. We therefore examine two possible remedies. The first is a multi-frame representation limited to Chomp, motivated by prior work on impartial games (Riis, 2024); the second is AlphaZero Auxiliary Loss (AZAL), which injects sparse oracle-derived supervision directly into training.

### 6.1 Multi-frame AlphaZero

The multi-frame variant does not resolve the main failure of vanilla AlphaZero. On the harder Chomp boards, its training curves in Figure 4 plateau at levels that are comparable to or worse than vanilla AlphaZero, especially in policy and total loss. The self-play traces in Tables 7a and 7c and the trajectories in Figure 3b–c reveal that multi-frame AlphaZero still fails to preserve the winning invariant reliably.

From Table 1, Multi-frame AlphaZero performs inferior to its vanilla counterpart. On $9 \times 10$, Multi-frame is significantly worse than Vanilla in full-game match (0.483 vs. 0.609), while on $10 \times 11$, it matches the performance of the baseline (0.463 vs. 0.474). Multi-frame also has shorter mean chains than Vanilla on both boards, with an average chain of 4.400 on $9 \times 10$ and 5.217 on $10 \times 11$.

Sampled-state evaluations for Multi-frame are subpar: 0.176 vs. 0.500 on $9 \times 10$, and 0.286 vs. 0.400 on $10 \times 11$. Therefore, the results indicate that adding additional frames fails to improve aggregate Chomp oracle consistency.

This negative result should not be interpreted as contradicting the Nim-specific motivation for multi-frame inputs (Riis, 2024). In Nim, consecutive positions expose simple local changes in heap sizes, making the relevant nimber-difference structure recoverable from a short history. Chomp differs because a move removes a two-dimensional set and induces nonlocal changes in the reachable space. Thus, a short stack of recent Chomp boards does not expose a simple local invariant analogous to the structure of Nim.

We evaluate multi-frame only on Chomp, so our multi-frame conclusion is restricted to the rectangular Chomp settings.

## 6.2   AZAL on Connect Four

AZAL improves Connect Four oracle consistency, especially by increasing oracle-match rate and delaying the first oracle mistake, but it does not reach perfect play.

**Trace evidence.**   In self-play, AZAL is markedly more stable than vanilla AlphaZero; see Table 8b and Figure 3a. The representative trace in Figure 3a depicts a cleaner early prefix for AZAL, while the aggregate results reveal how the mean first failure is delayed from 0.867 to 5.283.

The key difference from vanilla AlphaZero is that the number of decisive branch errors is substantially reduced and pushed later into the game. Even after the first major deviation, later play often remains locally strong within the new tactical regime.

From Tables 1 and 2, AZAL improves full-game match from 0.785 to 0.849, increases sampled-state match from 0.589 to 0.768, and delays the first failure from 0.867 to 5.283. However, AZAL does not improve mean longest chain (16.967 vs. Vanilla 17.000) and has zero perfect traces. Thus, AZAL improves the density and timing of oracle-consistent decisions in Connect Four, but does not produce fully oracle-consistent play.

**Training curves.**   The training curves in Figures 6a to 6d are consistent with this interpretation. Relative to vanilla AlphaZero, the auxiliary model maintains a lower policy loss for much of training, slightly lower value loss on average, and a steadily decreasing oracle-based auxiliary-policy loss. The gains are meaningful but limited: oscillations remain, and the model does not enter a fully stable low-noise convergence regime. Thus, AZAL strengthens the supervision signal enough to push the network closer to the oracle-preserving policy, but not enough to completely eliminate the flaws of its underlying self-play loop.

## 6.3   AZAL on Chomp

AZAL changes the Chomp results more dramatically than in Connect Four, with perfect full-game oracle consistency on $10 \times 11$ and high but incomplete consistency on $9 \times 10$.

**Trace evidence.**   On Chomp $10 \times 11$, AZAL achieves 60/60 perfect traces and a 1.000 (793/793) pooled oracle-match rate (Table 1). However, on Chomp $9 \times 10$, AZAL reaches 34/60 perfect traces and a 0.948 (1065/1123) match rate. On average, AZAL delays first failure on $9 \times 10$ to 11.923, while Vanilla and Multi-frame fail immediately on average.

Sampled-state evaluation reveal that AZAL performs best: 1.000 on $9 \times 10$ and 0.829 on $10 \times 11$. The sampled-state results on $10 \times 11$ illustrate that full-game perfection from the standard start does not imply perfect play over arbitrary sampled positions. Consequently, AZAL reaches near-complete recovery in Chomp, complete on $10 \times 11$ full-game traces, but not universal perfect play. A plausible explanation is that optimal gameplay on full traces renders AZAL brittle on arbitrary sampled positions that perfect play may never reach.

**Training curves second.**   The training curves match the trace-level improvement; see Figure 4. On both $9 \times 10$ and $10 \times 11$, AZAL lowers policy and total loss relative to both vanilla AlphaZero and the multi-frame baseline. The improvement is especially strong on $10 \times 11$, where policy loss falls well below the vanilla plateau and the value loss approaches near-zero by the end of training. The oracle-based $p_{\text{move}}$ loss also

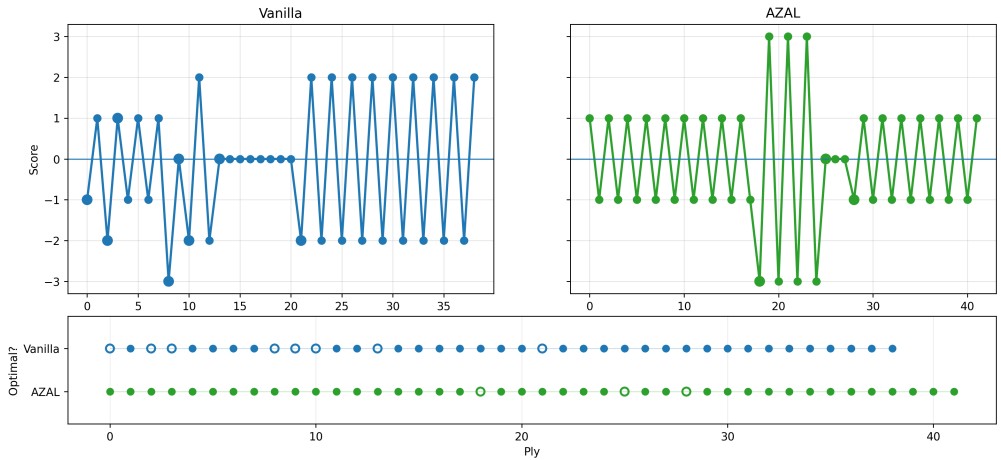

(a) Connect Four trace panel.

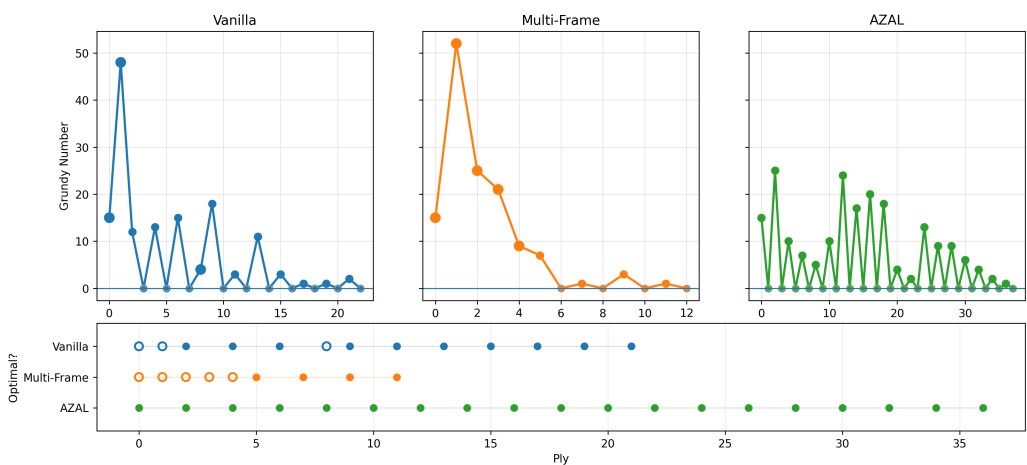

(b) Chomp 9×10 trace panel.

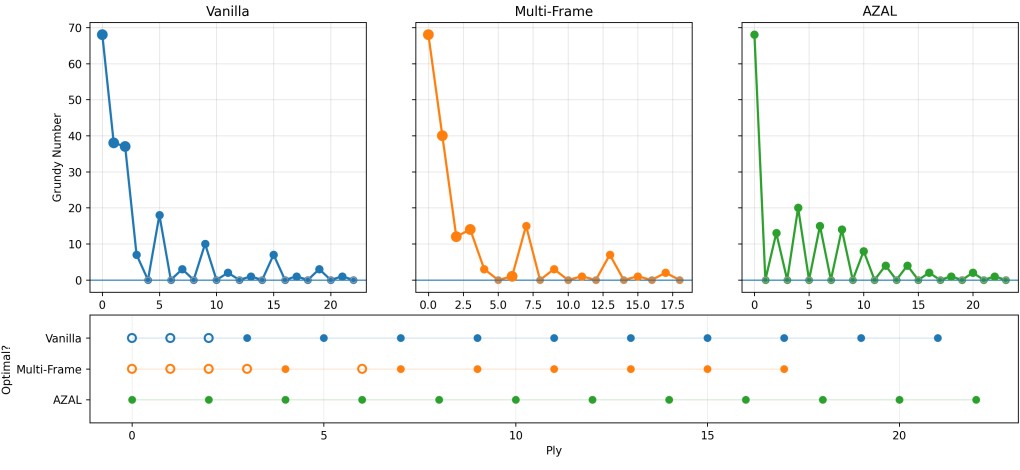

(c) Chomp 10×11 trace panel.

Figure 3: Deterministic greedy-rollout trace panels across games. Filled markers denote oracle-consistent moves; hollow markers denote non-oracle moves.

decreases substantially, providing direct evidence that the network is learning the invariant-preserving move structure.

## 6.4 Cross-domain comparison

AZAL helps in both domains, but more dramatically in Chomp than in Connect Four.

A first difference is the effective size and shape of the search target. Connect Four has a larger and deeper tactical space, with a branching factor up to seven throughout long prefixes of the game and many positions whose correct value depends on subtle long-horizon continuations. A single branch error can flip the position from winning to drawing or losing while leaving many legal moves still superficially plausible. Chomp, by contrast, often collapses much more sharply once an optimal move is chosen. Many optimal moves produce structurally simplified successor states by removing large portions of the board, making the consequences of correct play easier to internalize.

A second difference is the nature of the oracle signal itself. In Chomp, the main optimization target on the rectangular boards is especially crisp: from a winning position, Player 1 should move to $g = 0$, after which the opponent has no move that preserves a losing state for the other player. This makes the auxiliary signal highly selective and directly aligned with perfect play. In Connect Four, by contrast, both players remain strategically active and the oracle supervision must distinguish between subtly different long-horizon continuations, including choices among wins, draws, and delayed losses. The resulting supervision is still helpful, but less decisive.

Together, the aggregate results support the main claim of the paper. Better supervision closes much of the gap between strong play and oracle-consistent play, with the strongest gains in Chomp. However, the improvement is not uniform: AZAL is perfect on Chomp $10 \times 11$ full-game traces, high but incomplete on Chomp $9 \times 10$ and randomly sampled states, and still imperfect on Connect Four. Thus, the results support the interpretation that AlphaZero-style search-learning signals can be too weak to recover exact play reliably on their own, while also demonstrating that oracle-guided auxiliary supervision is not a complete solution in all settings.

# 7 Discussion

Table 3: Oracle value versus realized self-play outcome. The diagnostic counts positions where the oracle says the current player is losing, but the same player later wins the completed self-play rollout. For Chomp, oracle-losing means $g(s) = 0$; for Connect Four, oracle-losing means negative exact solver score.

| Game | Model | Positions | Oracle-losing states | Losing-but-won |
|---|---|---|---|---|
| Chomp $9\times10$ | Vanilla | 1287 | 487 | 0.031 (15/487) |
| | Multi-Frame | 835 | 272 | 0.022 (6/272) |
| | AZAL | 2188 | 1065 | 0.140 (149/1065) |
| Chomp $10\times11$ | Vanilla | 1505 | 484 | 0.043 (21/484) |
| | Multi-Frame | 1002 | 317 | 0.009 (3/317) |
| | AZAL | 1586 | 793 | 0.000 (0/793) |
| Connect Four | Vanilla | 2165 | 814 | 0.144 (117/814) |
| | AZAL | 2153 | 863 | 0.174 (150/863) |

The central conceptual result of this paper is that *superhuman play is not the same as perfect play.* Across both Connect Four and Chomp, vanilla AlphaZero attains strong empirical behavior, yet the trace analysis shows that decisive failures occur earlier, where the agent must preserve a long-horizon value invariant.

A remaining limitation is that our aggregate evaluation uses three seeds and a finite sampled-state set, not a dense distribution over all reachable positions. The deterministic traces are therefore used as illustrative case studies, while the main quantitative evidence comes from the multi-seed full-game and sampled-state evaluations. Future work should expand the sampled-state distribution, evaluate checkpoint sensitivity, and include larger board sizes and additional representation baselines. Furthermore, our results should not be

assumed to generalize beyond Connect Four and Chomp. The results are limited to two small oracle-evaluable games and should be interpreted as a diagnostic case study; broader claims regarding AlphaZero-style sparse-reward learning require additional games and larger settings. Finally, the results support, but do not fully isolate, the weak-search-learning-signal explanation; future work should vary search budget, architecture, $\lambda\_aux$, and checkpoint selection. We do not compare against Go-Exploit or other deep-state resampling methods; such comparisons would help separate data-distribution fixes from auxiliary-supervision fixes.

The results also suggest a specific failure. Standard AlphaZero self-play creates a moving-target problem: the network is asked to fit targets generated by its own evolving approximation, and early branch errors can corrupt the entire self-play distribution. In vanilla AlphaZero, the policy network is trained against visit-count targets generated by MCTS, while the value network is trained from terminal outcomes induced by self-play. Table 3 shows this failure mode occurs in our rollouts: some oracle-losing positions are later won by the same player because the opponent fails to exploit the advantage. In Connect Four, this occurs for both Vanilla and AZAL: 0.144 and 0.174. In Chomp $9 \times 10$, it also appears 14% of the time for AZAL. These results support the existence of self-play target corruption, but not a simple monotonic story where AZAL always reduces losing-but-won cases. The persistent oscillations in policy and value losses, together with the trace-level failures to maintain winning trajectories, are consistent with this diagnosis.

Connect Four and Chomp expose two versions of the same underlying issue. In Connect Four, perfect play is fragile: the game has a larger and deeper tactical space, draw states introduce value ambiguity, and small early deviations can shift the trajectory from winning to drawing or losing. As a result, even AZAL remains improved but imperfect. In Chomp, by contrast, the relevant structure is cleaner but more global: winning play is governed by the invariant $g(s) \neq 0 \rightarrow g(s') = 0$, and once this invariant is lost, perfect opposition makes recovery impossible. Here, vanilla AlphaZero fails sharply, but AZAL succeeds because the auxiliary signal aligns directly with the invariant that defines optimal play. The two domains therefore differ in surface form, but they point to the same deeper conclusion: the challenge is the reliable preservation of sparse long-horizon structure.

These observations sharpen the interpretation of AlphaZero-style success. AlphaZero can absolutely produce strong play, and in many domains that may be sufficient. But when the scientific question is whether the agent has recovered *exact* optimality, high win rates and smooth training curves are not enough. What matters is whether the search-learning loop consistently discovers, reinforces, and preserves the small set of decisions that define perfect play. Our results suggest that achieving this reliably may require stronger supervision, better search control, or both.

## 8 Conclusion

We studied the gap between strong play and perfect play in AlphaZero-style agents using Connect Four and Chomp as complementary oracle-evaluable domains. Vanilla AlphaZero achieves strong play in both games, but it does not preserve the exact trajectories required for optimal play. In Connect Four, it is better at tracking correct tactical continuations than at preserving the winning trajectory from the start. In Chomp, it fails to consistently maintain the $g = 0$ invariant that characterizes optimal winning play.

We then showed that stronger supervision materially changes this picture. Multi-frame inputs alone do not resolve the rectangular Chomp failure, while AZAL substantially improves oracle consistency across multi-seed full-game traces and sampled-state evaluations. AZAL reaches perfect full-game oracle consistency on Chomp $10 \times 11$, high but incomplete consistency on Chomp $9 \times 10$, and measurable but still imperfect improvement on Connect Four. These results suggest that weak search-derived supervision is a plausible limiting factor in AlphaZero-style recovery of exact play. Better supervision can close much of the gap, but the remaining challenge is to design search and training procedures that recover oracle-consistent play without relying on external oracles.

## Impact Statement

This paper studies the gap between strong play and perfect play in AlphaZero-style agents using Connect Four and Chomp as oracle-evaluable test domains. Its chief contribution is methodological: we show that high empirical performance can mask persistent failures to preserve the exact trajectories required for optimal play, and that stronger oracle-derived supervision can substantially reduce this gap. More broadly, this distinction between "superhuman" and "perfect" performance may matter beyond games, since learning systems in other domains can appear highly capable while still failing on rare but structurally important decisions. Potential negative impacts are indirect: methods that improve strategic decision making could in principle strengthen more general planning systems. However, our study is limited to small deterministic games with exact oracles and does not present a deployment method for real-world high-stakes applications.

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

# A   Oracle Implementation Details

## A.1   Connect Four oracle details

Our Connect Four oracle is the perfect solver of Pascal Pons (Pons, 2015), which evaluates positions via exact negamax alpha–beta search with transposition reuse. Actions are indexed by column, with illegal moves masked when a column is full. Internally, however, the solver uses a compact bitboard representation rather than an explicit two-dimensional array update. Although standard $6 \times 7$ Connect Four is solved (Allis, 1988), exact evaluation remains expensive in tactically delayed positions, where many legal continuations remain plausible and decisive structure appears deep in the tree. Under optimal play, a player on a winning state preserves positive-value control, while a player on a losing state selects the move with largest available score to delay defeat as long as possible.

## A.2   Chomp oracle details

Our Chomp oracle represents each state by its non-increasing row-length profile $\mathbf{s} = (s_0, \ldots, s_{H-1})$, where $s_r$ is the number of remaining squares in row $r$. A legal move selects a remaining square $(r, c)$ and truncates all rows at and below $r$ to length at most $c$:

$$s'_{r'} = \min(s_{r'}, c) \qquad \text{for all } r' \geq r.$$

Moves are indexed in row-major order by a single integer $n = rW + c$, providing a fixed-size action space with masking of illegal moves. The solver enumerates legal successors, recursively determines their Grundy numbers, and applies the mex rule with memoization. Even with memoization, the number of reachable profiles grows exponentially with board size, so runtime becomes restrictive on larger boards. The hardest states are typically dense configurations in which small local changes have long-horizon consequences.

# B   Additional Method Details

## B.1   Vanilla AlphaZero implementation

The vanilla baseline uses a convolutional residual policy–value network of the standard AlphaZero. Given an encoded board state $s$, the network outputs (i) policy logits over the fixed action space and (ii) a scalar value estimate:

$$(\mathbf{p}_\theta(s), v_\theta(s)) = f_\theta(s). \tag{5}$$

In our implementation, the input is a single board frame. The network begins with a $3 \times 3$ convolution, batch normalization, and ReLU activation, then passes through a stack of residual blocks with identity skip connections. The shared trunk is followed by separate policy and value heads. The policy head ends in a linear projection to the action space, while the value head ends in a scalar output with tanh activation so predictions lie in $[-1, 1]$.

More concretely, if $\mathbf{h}_0$ denotes the output of the initial convolutional block, then the residual trunk computes

$$\mathbf{h}_{k+1} = \mathbf{h}_k + F_k(\mathbf{h}_k), \tag{6}$$

where each $F_k$ consists of two $3 \times 3$ convolutions with batch normalization and ReLU nonlinearity. The policy head maps the final shared representation to logits over legal actions, and the value head maps it to a single scalar:

$$\mathbf{p}_\theta(s) \in \mathbb{R}^{|\mathcal{A}|}, \qquad v_\theta(s) \in [-1, 1]. \tag{7}$$

Search uses a PUCT-style selection rule,

$$a^* = \arg\max_a \left[ Q(s, a) + C\, P(a \mid s) \frac{\sqrt{\sum_b N(s, b)}}{1 + N(s, a)} \right], \tag{8}$$

where $Q(s, a)$ is the running value estimate, $N(s, a)$ is the visit count, and $P(a \mid s)$ is the policy prior returned by the network. Root priors are perturbed with Dirichlet noise during self-play, legal moves are masked before normalization, and the final root visit counts define the improved policy target.

Training data are generated by parallel self-play. For each visited state, the replay buffer stores the encoded state, the normalized MCTS visit-count distribution, and the final game outcome from the perspective of the player to move at that state. During optimization, the network is trained on minibatches sampled from this replay memory using the standard policy-plus-value loss

$$\mathcal{L}_{\text{AZ}} = \mathcal{L}_{\text{policy}} + \mathcal{L}_{\text{value}}, \tag{9}$$

where the policy term is cross-entropy against the MCTS-improved target distribution and the value term is mean-squared error against the terminal game outcome.

## B.2 Multi-frame AlphaZero implementation

The multi-frame AlphaZero model is included as a representation ablation. All search, replay, and optimization procedures are kept fixed so that the comparison isolates the effect of input representation alone.

The only substantive change relative to vanilla AlphaZero is the input encoding. Instead of supplying a single board snapshot $s_t$, the model receives a short history of recent board states,

$$(s_t, s_{t-1}, \ldots, s_{t-K+1}),$$

stacked along the channel dimension. In our implementation, the replay buffer stores encoded histories constructed from the three most recent frames of game state, rather than a single-frame board tensor.

The multi-frame network uses the same residual architecture as the vanilla baseline, with one change at the input layer: the first convolution accepts multiple input channels instead of one. In the implementation, the start block maps `num_frames` input channels into the shared hidden dimension, after which the same residual backbone, policy head, and value head are used as in the single-frame model. Thus, the multi-frame experiment should be interpreted as a controlled representation change, not as a different search procedure or optimization pipeline.

## B.3 AZAL-specific implementation

AZAL shares the same network architecture, self-play loop, MCTS procedure, and value targets as vanilla AlphaZero. The only implementation change is the addition of an auxiliary policy term:

$$\mathcal{L} = \mathcal{L}_{\text{policy}} + \mathcal{L}_{\text{value}} + \lambda_{\text{aux}} \mathcal{L}_{\text{aux}}, \tag{10}$$

where $\lambda_{\text{aux}}$ corresponds to the implementation parameter `p_move_lambda`. Domain-specific oracle-label construction for Connect Four and Chomp is described below.

**Connect Four auxiliary labels.** In Connect Four, oracle supervision comes from the perfect solver of Pascal Pons. During self-play, each replay element stores not only the encoded state, MCTS policy target, and final outcome, but also the move sequence leading to that state. After each self-play iteration, duplicated sequences are removed and the remaining unique sequences are queried in batches through the solver. The returned exact scores are then converted into oracle-best movesets and cached as replay-side labels before gradient updates begin. This means that oracle labels are attached once per iteration rather than recomputed inside every training batch.

**Chomp auxiliary labels.** In Chomp, oracle labels are derived from the Grundy solver. Positions that are terminal or fail oracle evaluation are masked out of the auxiliary term. To avoid repeated work, the set of best oracle moves is cached by serialized board state. As a result, the fraction of batch elements that contribute to the auxiliary loss can vary across training.

**Relationship to vanilla AlphaZero.** AZAL is therefore a slight modification of the standard AlphaZero loop. It does not replace MCTS-based policy improvement, alter value targets, or inject oracle information into search itself. Instead, it sharpens the learning signal at training time by explicitly favoring moves known to be optimal under an exact oracle, while leaving the rest of the self-play pipeline unchanged.

### B.4 Protocol and logging details

All experiments use the same high-level training loop. At each iteration, the current network is placed in evaluation mode and used to guide MCTS-based self-play across multiple games in parallel. For each root state, MCTS returns visit counts over legal actions, these counts are normalized to produce an improved policy target, and an action is sampled from a temperature-adjusted distribution for rollout generation. Once a game terminates, every stored state in that trajectory is assigned the final outcome from the perspective of the player to move at that state and added to replay memory.

For vanilla and multi-frame AlphaZero, the total loss is the sum of policy and value losses. In practice, we log these components separately as `policy_loss` and `value_loss`, along with their sum `total_loss`. For AZAL, we additionally log the auxiliary policy term `p_move_loss` and the fraction of batch elements for which oracle supervision is available, `p_move_labeled_frac`. In Connect Four, we also track oracle-query statistics per iteration, including the total number of queried positions and the number of unique positions after deduplication.

We evaluate each method against exact or oracle-derived notions of optimality rather than relying only on empirical win rate. In Connect Four, we compare play against the exact game-theoretic scores returned by the perfect solver. In Chomp, we use the Grundy oracle to classify positions as winning or losing and to determine if a move preserves the invariant $g(s) \neq 0 \rightarrow g(s') = 0$ that characterizes optimal first-player control.

To make these comparisons explicit, we log move-by-move self-play traces and evaluate each move with oracle annotations. In Chomp, for each ply $t$ we record the played action, the oracle Grundy number $g(s_t)$ of the state before the move, and an oracle reference best move when $g(s_t) \neq 0$. In Connect Four, we log the played column together with the oracle score of the current position and the solver's optimal moveset for that state.

## C  Additional Results

Appendix C reports player-specific oracle-match rates for the aggregate full-game traces. These results separate first-player and second-player labeled decisions, which is important in Chomp as positions with Grundy number zero have no oracle-best move. Thus, N/A entries indicate the absence of labeled oracle decisions.

Table 4: Player-specific oracle-match rates across full-game self-play traces. $Match_1$ is computed over labeled first-player plies and $Match_2$ over labeled second-player plies. In Chomp, states with Grundy number zero have no oracle-best move; therefore, some player-specific entries are N/A as no labeled oracle decision is defined.

| Game | Model | Match | $Match_1$ | $Match_2$ |
|---|---|---|---|---|
| Chomp $9 \times 10$ | Vanilla | 0.609 (487/800) | 0.578 (236/408) | 0.640 (251/392) |
| | Multi-Frame | 0.483 (272/563) | 0.456 (136/298) | 0.513 (136/265) |
| | AZAL | 0.948 (1065/1123) | 0.962 (861/895) | 0.895 (204/228) |
| Chomp $10 \times 11$ | Vanilla | 0.474 (484/1021) | 0.416 (204/490) | 0.527 (280/531) |
| | Multi-Frame | 0.463 (317/685) | 0.458 (168/367) | 0.469 (149/318) |
| | AZAL | 1.000 (793/793) | 1.000 (793/793) | N/A |
| Connect Four | Vanilla | 0.785 (1700/2165) | 0.771 (843/1094) | 0.800 (857/1071) |
| | AZAL | 0.849 (1828/2153) | 0.840 (913/1087) | 0.858 (915/1066) |

# D   Hyperparameter Table

Appendix D summarizes the main reproducibility details for all aggregate experiments. These settings specify the architecture, search budget, self-play volume, optimization parameters, auxiliary-loss weight, exploration schedule, checkpoint rule, and number of seeds used in the reported results.

Table 5: Reproducibility details for the main experiments. Res. denotes residual blocks/channels. MCTS denotes simulations per move. Iter. denotes training iterations. Games/Iter. denotes self-play games per training iteration. Buffer indicates replay-buffer contents. Temp. denotes the self-play temperature schedule. Noise denotes root Dirichlet noise. Ckpt. denotes the checkpoint-selection rule. Seeds are reported as training seeds / evaluation seeds. Multi-Frame was evaluated only on the rectangular Chomp boards, so no Connect Four Multi-Frame row is reported.

| Game | Model | Board | Arch. | Res. | MCTS | Iter. | Games/Iter. | Buffer | Batch | LR | $\lambda_{aux}$ | Temp. | Noise | Ckpt. | Seeds |
|------|-------|-------|-------|------|------|-------|-------------|--------|-------|-----|------|-------|-------|-------|-------|
| | Vanilla | 9×10 | Conv-ResNet, 1-plane | 9/128 | 800 | 10 | 700 | all states / 700 games | 128 | 0.001 | N/A | const. 1.25 | $\epsilon = 0.25, \alpha = 0.3$ | last iter. | 0,1,2 / 100000–100002 |
| | Vanilla | 10×11 | Conv-ResNet, 1-plane | 9/128 | 800 | 10 | 700 | all states / 700 games | 128 | 0.001 | N/A | const. 1.25 | $\epsilon = 0.25, \alpha = 0.3$ | last iter. | 0,1,2 / 100000–100002 |
| Chomp | Multi-Frame | 9×10 | Conv-ResNet, 3-frame | 9/128 | 800 | 10 | 700 | all states / 700 games | 128 | 0.001 | N/A | const. 1.25 | $\epsilon = 0.25, \alpha = 0.3$ | last iter. | 0,1,2 / 100000–100002 |
| | Multi-Frame | 10×11 | Conv-ResNet, 3-frame | 9/128 | 800 | 10 | 700 | all states / 700 games | 128 | 0.001 | N/A | const. 1.25 | $\epsilon = 0.25, \alpha = 0.3$ | last iter. | 0,1,2 / 100000–100002 |
| | AZAL | 9×10 | Conv-ResNet, 1-plane | 9/128 | 800 | 10 | 700 | all states / 700 games | 128 | 0.001 | 1.0 | const. 1.25 | $\epsilon = 0.25, \alpha = 0.3$ | last iter. | 0,1,2 / 100000–100002 |
| | AZAL | 10×11 | Conv-ResNet, 1-plane | 9/128 | 800 | 10 | 700 | all states / 700 games | 128 | 0.001 | 1.0 | const. 1.25 | $\epsilon = 0.25, \alpha = 0.3$ | last iter. | 0,1,2 / 100000–100002 |
| Connect Four | Vanilla | 6×7 | Conv-ResNet, 3-plane | 9/128 | 1000 | 20 | 700 | all states / 700 games | 128 | 0.001 | N/A | const. 1.25 | $\epsilon = 0.25, \alpha = 0.3$ | last iter. | 0,1,2 / 100000–100002 |
| | AZAL | 6×7 | Conv-ResNet, 3-plane | 9/128 | 1000 | 20 | 700 | all states / 700 games | 128 | 0.001 | 1.0 | const. 1.25 | $\epsilon = 0.25, \alpha = 0.3$ | last iter. | 0,1,2 / 100000–100002 |

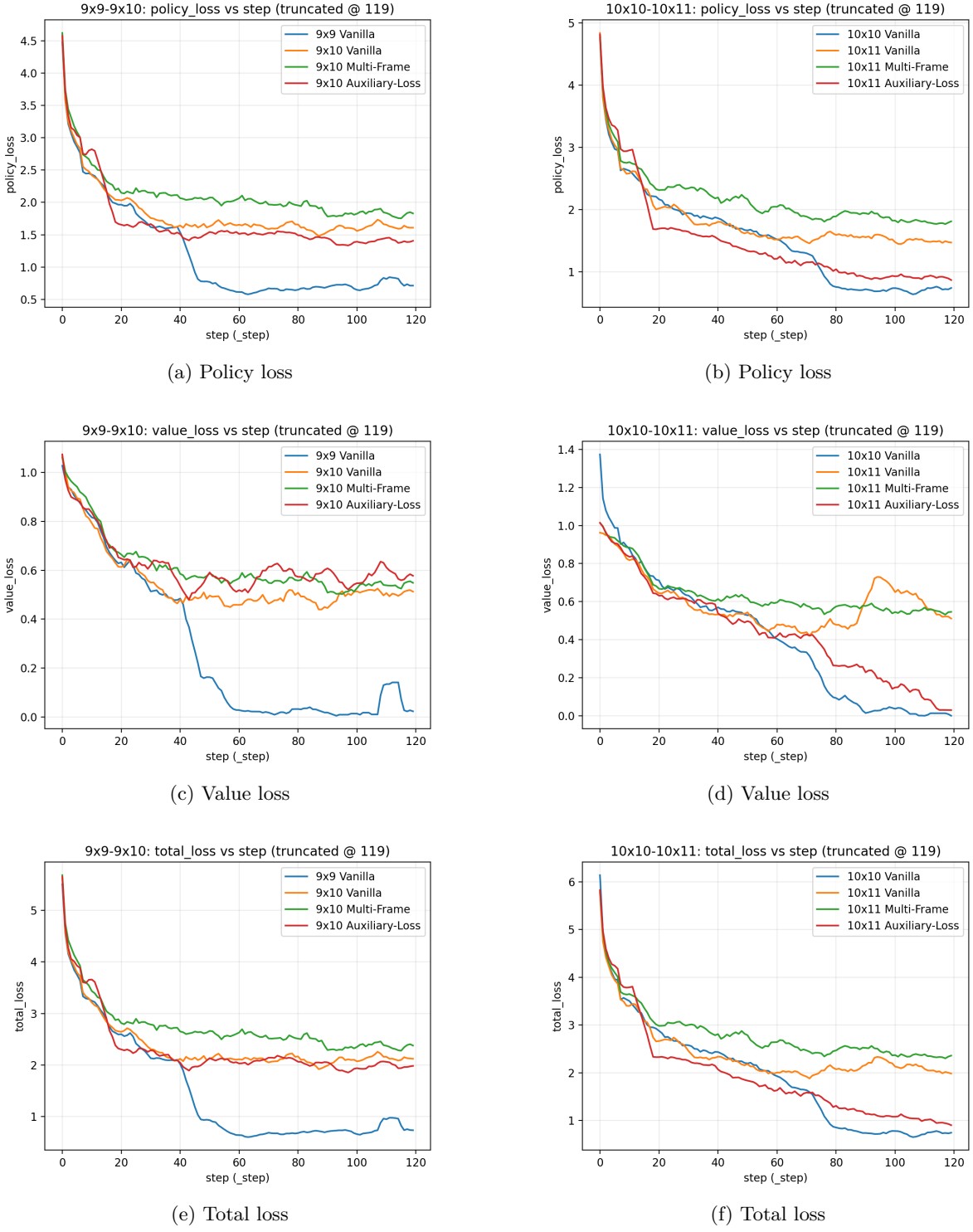

(a) Policy loss

(b) Policy loss

(c) Value loss

(d) Value loss

(e) Total loss

(f) Total loss

Figure 4: Smoothed training losses for Chomp on the $9 \times 9$, $9 \times 10$, $10 \times 10$, and $10 \times 11$ boards across model variants. The left column shows results for the $9 \times 9$ and $9 \times 10$ boards, and the right column shows results for the $10 \times 10$ and $10 \times 11$ boards. The rows report policy loss, value loss, and total loss, where total loss is the sum of the policy and value terms. Curves are truncated to a common training horizon within each board-size group to enable direct comparison of convergence trends.

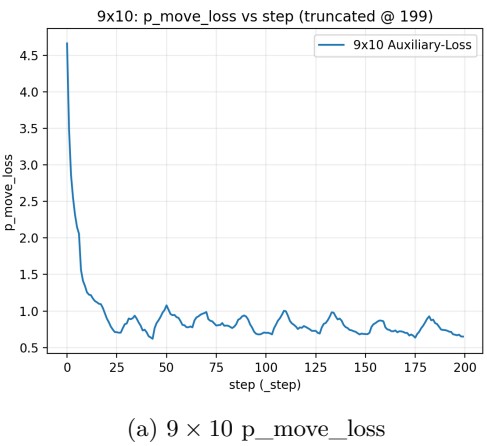
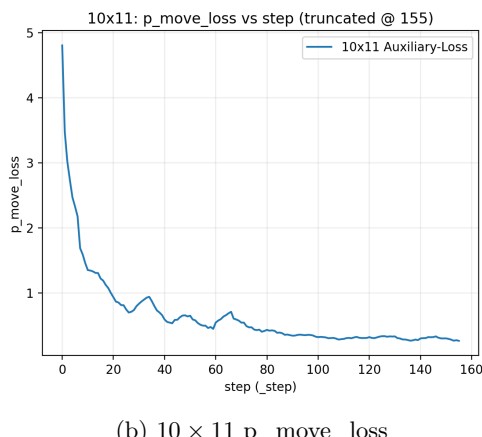

(a) $9 \times 10$ p_move_loss

(b) $10 \times 11$ p_move_loss

Figure 5: Smoothed auxiliary-policy loss $p_{\text{move}}$ for AlphaZero-Auxiliary Loss on Chomp for the $9 \times 10$ and $10 \times 11$ boards. Curves are truncated to a common training horizon within each board-size group to enable direct comparison of convergence trends.

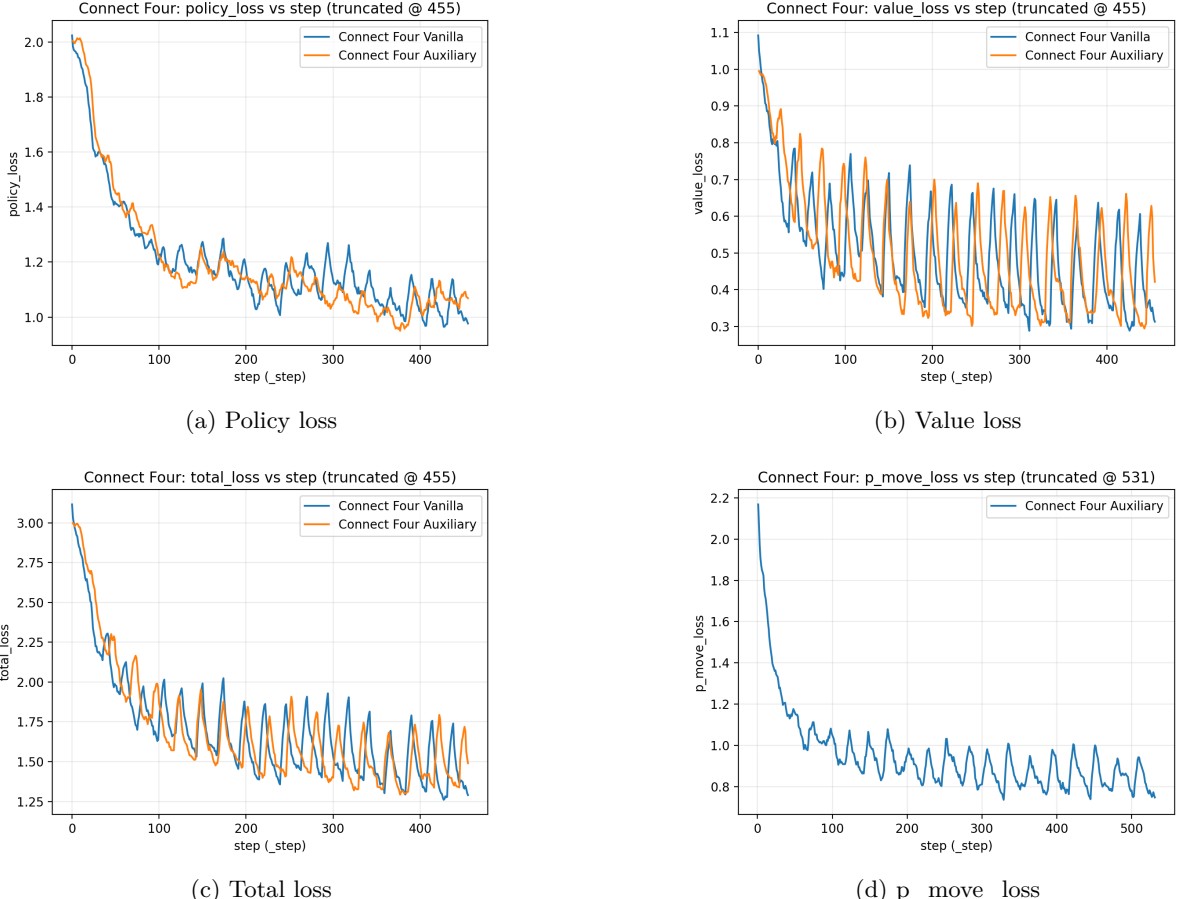

(a) Policy loss

(b) Value loss

(c) Total loss

(d) p_move_loss

Figure 6: Smoothed training losses for Connect Four across model variants. Curves are truncated to a common training horizon to enable direct comparison of convergence trends. The panels report policy loss, value loss, total loss, and, for AlphaZero-Auxiliary Loss, the auxiliary-policy loss $p_{\text{move}}$.

Table 6: Move-level trace tables for vanilla AlphaZero on Chomp in self-play across board sizes. Here $g(s_t)$ denotes the Grundy number of the position before the move at ply $t$. When $g(s_t) = 0$, the position is losing, so the oracle does not define a "best" move; these entries are shown as $-$.

| Ply $t$ | Player | $g(s_t)$ | AZ move $n$ | Oracle best $n$ | Optimal? |
|---|---|---|---|---|---|
| 0 | AZ-1 | 20 | 10 | 10 | yes |
| 1 | AZ-2 | 0 | 72 | – | – |
| 2 | AZ-1 | 15 | 8 | 8 | yes |
| 3 | AZ-2 | 0 | 63 | – | – |
| 4 | AZ-1 | 1 | 7 | 7 | yes |
| 5 | AZ-2 | 0 | 54 | – | – |
| 6 | AZ-1 | 3 | 6 | 6 | yes |
| 7 | AZ-2 | 0 | 5 | – | – |
| 8 | AZ-1 | 1 | 45 | 45 | yes |
| 9 | AZ-2 | 0 | 36 | – | – |
| 10 | AZ-1 | 7 | 4 | 4 | yes |
| 11 | AZ-2 | 0 | 3 | – | – |
| 12 | AZ-1 | 1 | 27 | 27 | yes |
| 13 | AZ-2 | 0 | 18 | – | – |
| 14 | AZ-1 | 3 | 2 | 2 | yes |
| 15 | AZ-2 | 0 | 1 | – | – |
| 16 | AZ-1 | 1 | 9 | 9 | yes |
| 17 | AZ-2 | 0 | 0 | – | – |

(a) Chomp Vanilla AZ ($9 \times 9$)

| Ply $t$ | Player | $g(s_t)$ | AZ move $n$ | Oracle best $n$ | Optimal? |
|---|---|---|---|---|---|
| 0 | AZ-1 | 15 | 72 | 67, 84 | no |
| 1 | AZ-2 | 48 | 24 | 46 | no |
| 2 | AZ-1 | 12 | 52 | 33, 52, 60 | yes |
| 3 | AZ-2 | 0 | 60 | – | – |
| 4 | AZ-1 | 13 | 33 | 8, 17, 33 | yes |
| 5 | AZ-2 | 0 | 18 | – | – |
| 6 | AZ-1 | 15 | 23 | 23 | yes |
| 7 | AZ-2 | 0 | 8 | – | – |
| 8 | AZ-1 | 4 | 42 | 16 | no |
| 9 | AZ-2 | 18 | 14 | 14 | yes |
| 10 | AZ-1 | 0 | 32 | – | – |
| 11 | AZ-2 | 3 | 13 | 13 | yes |
| 12 | AZ-1 | 0 | 22 | – | – |
| 13 | AZ-2 | 11 | 50 | 50 | yes |
| 14 | AZ-1 | 0 | 5 | – | – |
| 15 | AZ-2 | 3 | 30 | 11, 30 | yes |
| 16 | AZ-1 | 0 | 12 | – | – |
| 17 | AZ-2 | 1 | 4 | 4 | yes |
| 18 | AZ-1 | 0 | 11 | – | – |
| 19 | AZ-2 | 1 | 3 | 3 | yes |
| 20 | AZ-1 | 0 | 10 | – | – |
| 21 | AZ-2 | 2 | 1 | 1 | yes |
| 22 | AZ-1 | 0 | 0 | – | – |

(b) Chomp Vanilla AZ ($9 \times 10$)

| Ply $t$ | Player | $g(s_t)$ | AZ move $n$ | Oracle best $n$ | Optimal? |
|---|---|---|---|---|---|
| 0 | AZ-1 | 19 | 11 | 11 | yes |
| 1 | AZ-2 | 0 | 90 | – | – |
| 2 | AZ-1 | 1 | 9 | 9 | yes |
| 3 | AZ-2 | 0 | 80 | – | – |
| 4 | AZ-1 | 15 | 8 | 8 | yes |
| 5 | AZ-2 | 0 | 70 | – | – |
| 6 | AZ-1 | 1 | 7 | 7 | yes |
| 7 | AZ-2 | 0 | 6 | – | – |
| 8 | AZ-1 | 3 | 60 | 60 | yes |
| 9 | AZ-2 | 0 | 5 | – | – |
| 10 | AZ-1 | 1 | 50 | 50 | yes |
| 11 | AZ-2 | 0 | 4 | – | – |
| 12 | AZ-1 | 7 | 40 | 40 | yes |
| 13 | AZ-2 | 0 | 3 | – | – |
| 14 | AZ-1 | 1 | 30 | 30 | yes |
| 15 | AZ-2 | 0 | 2 | – | – |
| 16 | AZ-1 | 3 | 20 | 20 | yes |
| 17 | AZ-2 | 0 | 1 | – | – |
| 18 | AZ-1 | 1 | 10 | 10 | yes |
| 19 | AZ-2 | 0 | 0 | – | – |

(c) Chomp Vanilla AZ ($10 \times 10$)

| Ply $t$ | Player | $g(s_t)$ | AZ move $n$ | Oracle best $n$ | Optimal? |
|---|---|---|---|---|---|
| 0 | AZ-1 | 68 | 6 | 24 | no |
| 1 | AZ-2 | 38 | 93 | 46 | no |
| 2 | AZ-1 | 37 | 35 | 46 | no |
| 3 | AZ-2 | 7 | 88 | 88 | yes |
| 4 | AZ-1 | 0 | 26 | – | – |
| 5 | AZ-2 | 18 | 45 | 45 | yes |
| 6 | AZ-1 | 0 | 25 | – | – |
| 7 | AZ-2 | 3 | 34 | 34 | yes |
| 8 | AZ-1 | 0 | 15 | – | – |
| 9 | AZ-2 | 10 | 23 | 23, 77 | yes |
| 10 | AZ-1 | 0 | 12 | – | – |
| 11 | AZ-2 | 2 | 66 | 66 | yes |
| 12 | AZ-1 | 0 | 55 | – | – |
| 13 | AZ-2 | 1 | 5 | 5 | yes |
| 14 | AZ-1 | 0 | 4 | – | – |
| 15 | AZ-2 | 7 | 44 | 44 | yes |
| 16 | AZ-1 | 0 | 33 | – | – |
| 17 | AZ-2 | 1 | 3 | 3 | yes |
| 18 | AZ-1 | 0 | 22 | – | – |
| 19 | AZ-2 | 3 | 2 | 2 | yes |
| 20 | AZ-1 | 0 | 1 | – | – |
| 21 | AZ-2 | 1 | 11 | 11 | yes |
| 22 | AZ-1 | 0 | 0 | – | – |

(d) Chomp Vanilla AZ ($10 \times 11$)

Table 7: Move-level trace tables for Multi-Frame AlphaZero and AlphaZero-Auxiliary Loss on Chomp in self-play across board sizes. Here $g(s_t)$ denotes the Grundy number of the position before the move at ply $t$. When $g(s_t) = 0$, the position is losing, so the oracle does not define a "best" move; these entries are shown as –.

| Ply $t$ | Player | $g(s_t)$ | AZ move $n$ | Oracle best $n$ | Optimal? |
|---|---|---|---|---|---|
| 0 | AZ-1 | 15 | 48 | 67, 84 | no |
| 1 | AZ-2 | 52 | 46 | 66 | no |
| 2 | AZ-1 | 25 | 30 | 54, 72 | no |
| 3 | AZ-2 | 21 | 4 | 15 | no |
| 4 | AZ-1 | 9 | 22 | 12 | no |
| 5 | AZ-2 | 7 | 12 | 12 | yes |
| 6 | AZ-1 | 0 | 11 | – | – |
| 7 | AZ-2 | 1 | 3 | 3 | yes |
| 8 | AZ-1 | 0 | 20 | – | – |
| 9 | AZ-2 | 3 | 2 | 2 | yes |
| 10 | AZ-1 | 0 | 1 | – | – |
| 11 | AZ-2 | 1 | 10 | 10 | yes |
| 12 | AZ-1 | 0 | 0 | – | – |

(a) Chomp Multi-Frame AZ ($9 \times 10$)

| Ply $t$ | Player | $g(s_t)$ | AZ move $n$ | Oracle best $n$ | Optimal? |
|---|---|---|---|---|---|
| 0 | AZ-1 | 15 | 84 | 67, 84 | yes |
| 1 | AZ-2 | 0 | 55 | – | – |
| 2 | AZ-1 | 25 | 74 | 18, 36, 74 | yes |
| 3 | AZ-2 | 0 | 49 | – | – |
| 4 | AZ-1 | 10 | 83 | 83 | yes |
| 5 | AZ-2 | 0 | 46 | – | – |
| 6 | AZ-1 | 7 | 18 | 18 | yes |
| 7 | AZ-2 | 0 | 54 | – | – |
| 8 | AZ-1 | 5 | 82 | 35, 82 | yes |
| 9 | AZ-2 | 0 | 73 | – | – |
| 10 | AZ-1 | 10 | 35 | 35 | yes |
| 11 | AZ-2 | 0 | 60 | – | – |
| 12 | AZ-1 | 24 | 7 | 7 | yes |
| 13 | AZ-2 | 0 | 44 | – | – |
| 14 | AZ-1 | 17 | 53 | 53 | yes |
| 15 | AZ-2 | 0 | 52 | – | – |
| 16 | AZ-1 | 20 | 34 | 34 | yes |
| 17 | AZ-2 | 0 | 43 | – | – |
| 18 | AZ-1 | 18 | 26 | 26 | yes |
| 19 | AZ-2 | 0 | 25 | – | – |
| 20 | AZ-1 | 4 | 41 | 41 | yes |
| 21 | AZ-2 | 0 | 5 | – | – |
| 22 | AZ-1 | 2 | 33 | 33 | yes |
| 23 | AZ-2 | 0 | 32 | – | – |
| 24 | AZ-1 | 13 | 24 | 24 | yes |
| 25 | AZ-2 | 0 | 31 | – | – |
| 26 | AZ-1 | 9 | 3 | 3, 12 | yes |
| 27 | AZ-2 | 0 | 22 | – | – |
| 28 | AZ-1 | 9 | 50 | 50 | yes |
| 29 | AZ-2 | 0 | 2 | – | – |
| 30 | AZ-1 | 6 | 40 | 40 | yes |
| 31 | AZ-2 | 0 | 21 | – | – |
| 32 | AZ-1 | 4 | 30 | 30 | yes |
| 33 | AZ-2 | 0 | 20 | – | – |
| 34 | AZ-1 | 2 | 11 | 11 | yes |
| 35 | AZ-2 | 0 | 10 | – | – |
| 36 | AZ-1 | 1 | 1 | 1 | yes |
| 37 | AZ-2 | 0 | 0 | – | – |

(b) Chomp Auxiliary Loss AZ ($9 \times 10$)

| Ply $t$ | Player | $g(s_t)$ | AZ move $n$ | Oracle best $n$ | Optimal? |
|---|---|---|---|---|---|
| 0 | AZ-1 | 68 | 70 | 24 | no |
| 1 | AZ-2 | 40 | 100 | 24 | no |
| 2 | AZ-1 | 12 | 25 | 2 | no |
| 3 | AZ-2 | 14 | 12 | 2 | no |
| 4 | AZ-1 | 3 | 10 | 10 | yes |
| 5 | AZ-2 | 0 | 9 | – | – |
| 6 | AZ-1 | 1 | 7 | 99 | no |
| 7 | AZ-2 | 15 | 77 | 77 | yes |
| 8 | AZ-1 | 0 | 66 | – | – |
| 9 | AZ-2 | 3 | 6 | 6 | yes |
| 10 | AZ-1 | 0 | 55 | – | – |
| 11 | AZ-2 | 1 | 5 | 5 | yes |
| 12 | AZ-1 | 0 | 4 | – | – |
| 13 | AZ-2 | 7 | 44 | 44 | yes |
| 14 | AZ-1 | 0 | 33 | – | – |
| 15 | AZ-2 | 1 | 3 | 3 | yes |
| 16 | AZ-1 | 0 | 11 | – | – |
| 17 | AZ-2 | 2 | 1 | 1 | yes |
| 18 | AZ-1 | 0 | 0 | – | – |

(c) Chomp Multi-Frame AZ ($10 \times 11$)

| Ply $t$ | Player | $g(s_t)$ | AZ move $n$ | Oracle best $n$ | Optimal? |
|---|---|---|---|---|---|
| 0 | AZ-1 | 68 | 24 | 24 | yes |
| 1 | AZ-2 | 0 | 19 | – | – |
| 2 | AZ-1 | 13 | 99 | 99 | yes |
| 3 | AZ-2 | 0 | 18 | – | – |
| 4 | AZ-1 | 20 | 67 | 10, 67 | yes |
| 5 | AZ-2 | 0 | 10 | – | – |
| 6 | AZ-1 | 15 | 17 | 17 | yes |
| 7 | AZ-2 | 0 | 56 | – | – |
| 8 | AZ-1 | 14 | 15 | 15 | yes |
| 9 | AZ-2 | 0 | 88 | – | – |
| 10 | AZ-1 | 8 | 7 | 7, 14, 34 | yes |
| 11 | AZ-2 | 0 | 34 | – | – |
| 12 | AZ-1 | 4 | 13 | 13 | yes |
| 13 | AZ-2 | 0 | 66 | – | – |
| 14 | AZ-1 | 4 | 23 | 5, 23 | yes |
| 15 | AZ-2 | 0 | 4 | – | – |
| 16 | AZ-1 | 2 | 55 | 55 | yes |
| 17 | AZ-2 | 0 | 44 | – | – |
| 18 | AZ-1 | 1 | 12 | 12 | yes |
| 19 | AZ-2 | 0 | 2 | – | – |
| 20 | AZ-1 | 2 | 22 | 22 | yes |
| 21 | AZ-2 | 0 | 11 | – | – |
| 22 | AZ-1 | 1 | 1 | 1 | yes |
| 23 | AZ-2 | 0 | 0 | – | – |

(d) Chomp Auxiliary Loss AZ ($10 \times 11$)

Table 8: Move-level trace tables for vanilla AlphaZero and AlphaZero-Auxiliary Loss on Connect Four in self-play. Here $\sigma(s_t)$ denotes the oracle score associated with the action chosen at ply $t$ from state $s_t$. Positive values indicate winning continuations, negative values indicate losing continuations, and zero indicates drawing continuations.

| Ply $t$ | Player | $\sigma(s_t)$ | AZ move $n$ | Oracle best $n$ | Optimal? |
|---|---|---|---|---|---|
| 0 | AZ-1 | -1 | 5 | 3 | no |
| 1 | AZ-2 | 1 | 4 | 4 | yes |
| 2 | AZ-1 | -2 | 2 | 4, 5 | no |
| 3 | AZ-2 | 1 | 2 | 4 | no |
| 4 | AZ-1 | -1 | 4 | 4 | yes |
| 5 | AZ-2 | 1 | 4 | 4 | yes |
| 6 | AZ-1 | -1 | 4 | 4 | yes |
| 7 | AZ-2 | 1 | 4 | 4, 5 | yes |
| 8 | AZ-1 | -3 | 4 | 3, 2, 5, 0, 6 | no |
| 9 | AZ-2 | 0 | 0 | 3 | no |
| 10 | AZ-1 | -2 | 2 | 0 | no |
| 11 | AZ-2 | 2 | 2 | 2 | yes |
| 12 | AZ-1 | -2 | 2 | 3, 2, 0, 6 | yes |
| 13 | AZ-2 | 0 | 2 | 5 | no |
| 14 | AZ-1 | 0 | 0 | 3, 0 | yes |
| 15 | AZ-2 | 0 | 6 | 3, 1, 5, 0, 6 | yes |
| 16 | AZ-1 | 0 | 0 | 3, 5, 0 | yes |
| 17 | AZ-2 | 0 | 0 | 3, 1, 5, 0, 6 | yes |
| 18 | AZ-1 | 0 | 0 | 3, 1, 5, 0, 6 | yes |
| 19 | AZ-2 | 0 | 0 | 3, 1, 5, 0, 6 | yes |
| 20 | AZ-1 | 0 | 3 | 3, 1, 5, 6 | yes |
| 21 | AZ-2 | -2 | 3 | 1, 5, 6 | no |
| 22 | AZ-1 | 2 | 3 | 3 | yes |
| 23 | AZ-2 | -2 | 3 | 3 | yes |
| 24 | AZ-1 | 2 | 5 | 5 | yes |
| 25 | AZ-2 | -2 | 5 | 5 | yes |
| 26 | AZ-1 | 2 | 3 | 3, 5 | yes |
| 27 | AZ-2 | -2 | 3 | 3, 5, 6 | yes |
| 28 | AZ-1 | 2 | 5 | 5 | yes |
| 29 | AZ-2 | -2 | 5 | 5 | yes |
| 30 | AZ-1 | 2 | 5 | 1, 5, 6 | yes |
| 31 | AZ-2 | -2 | 6 | 1, 6 | yes |
| 32 | AZ-1 | 2 | 1 | 1, 6 | yes |
| 33 | AZ-2 | -2 | 6 | 6 | yes |
| 34 | AZ-1 | 2 | 6 | 6 | yes |
| 35 | AZ-2 | -2 | 6 | 6 | yes |
| 36 | AZ-1 | 2 | 6 | 6 | yes |
| 37 | AZ-2 | -2 | 1 | 1 | yes |
| 38 | AZ-1 | 2 | 1 | 1 | yes |

(a) Connect Four Vanilla AZ

| Ply $t$ | Player | $\sigma(s_t)$ | AZ move $n$ | Oracle best $n$ | Optimal? |
|---|---|---|---|---|---|
| 0 | AZ-1 | 1 | 3 | 3 | yes |
| 1 | AZ-2 | -1 | 3 | 3 | yes |
| 2 | AZ-1 | 1 | 3 | 3 | yes |
| 3 | AZ-2 | -1 | 3 | 3 | yes |
| 4 | AZ-1 | 1 | 3 | 3 | yes |
| 5 | AZ-2 | -1 | 2 | 3, 2, 4, 1, 5, 0, 6 | yes |
| 6 | AZ-1 | 1 | 2 | 2, 5 | yes |
| 7 | AZ-2 | -1 | 2 | 2 | yes |
| 8 | AZ-1 | 1 | 2 | 2 | yes |
| 9 | AZ-2 | -1 | 0 | 2, 5, 0 | yes |
| 10 | AZ-1 | 1 | 5 | 3, 2, 5 | yes |
| 11 | AZ-2 | -1 | 6 | 6 | yes |
| 12 | AZ-1 | 1 | 4 | 3, 2, 4, 5, 0, 6 | yes |
| 13 | AZ-2 | -1 | 4 | 4 | yes |
| 14 | AZ-1 | 1 | 4 | 4 | yes |
| 15 | AZ-2 | -1 | 4 | 4, 5, 6 | yes |
| 16 | AZ-1 | 1 | 4 | 4 | yes |
| 17 | AZ-2 | -1 | 2 | 3, 2, 4, 1, 5, 0, 6 | yes |
| 18 | AZ-1 | -3 | 6 | 5 | no |
| 19 | AZ-2 | 3 | 6 | 6 | yes |
| 20 | AZ-1 | -3 | 6 | 6 | yes |
| 21 | AZ-2 | 3 | 6 | 6 | yes |
| 22 | AZ-1 | -3 | 4 | 3, 2, 4, 5, 0, 6 | yes |
| 23 | AZ-2 | 3 | 3 | 3, 2, 5, 0, 6 | yes |
| 24 | AZ-1 | -3 | 6 | 2, 5, 0, 6 | yes |
| 25 | AZ-2 | 0 | 1 | 2, 5, 0 | no |
| 26 | AZ-1 | 0 | 1 | 1 | yes |
| 27 | AZ-2 | 0 | 1 | 1 | yes |
| 28 | AZ-1 | -1 | 1 | 2 | no |
| 29 | AZ-2 | 1 | 2 | 2 | yes |
| 30 | AZ-1 | -1 | 0 | 1, 5, 0 | yes |
| 31 | AZ-2 | 1 | 5 | 5, 0 | yes |
| 32 | AZ-1 | -1 | 5 | 1, 5, 0 | yes |
| 33 | AZ-2 | 1 | 5 | 5, 0 | yes |
| 34 | AZ-1 | -1 | 0 | 1, 5, 0 | yes |
| 35 | AZ-2 | 1 | 0 | 0 | yes |
| 36 | AZ-1 | -1 | 1 | 1, 5, 0 | yes |
| 37 | AZ-2 | 1 | 1 | 1 | yes |
| 38 | AZ-1 | -1 | 5 | 5 | yes |
| 39 | AZ-2 | 1 | 5 | 5 | yes |
| 40 | AZ-1 | -1 | 0 | 0 | yes |
| 41 | AZ-2 | 1 | 0 | 0 | yes |

(b) Connect Four Auxiliary Loss AZ

