# OpenReview forum: "AlphaZero in Sparsely Rewarded Games: Limits and Auxiliary Supervision"
_TMLR — Decision pending for TMLR_

### Review · Reviewer_1oxy · 2026-06-10

**Summary Of Contributions:**

This paper studies whether AlphaZero-style self-play recovers optimal play in addition to strong play in two oracle-evaluable games: Connect Four and Chomp. Under the self-play + Monte-Carlo tree search pipeline, the paper compares three algorithms: Vanilla AlphaZero, multi-frame variant of AlphaZero, and the AlphaZero Auxiliary Loss (AZAL) variant. The authors find that vanilla AlphaZero achieves strong play but not optimal play. The multi-frame variant does not fix the gap, whereas the AZAL variant reaches perfect play on Chomp and near-perfect play on Connect Four.

**Audience:**

Yes

**Audience Explanation:**

Yes. The distinction between strong and perfect play, made concrete through oracle-evaluable games, is a clean idea, and readers working on AlphaZero, combinatorial games, and RL self-play dynamics would find the framing useful.

**Broader Impact Concerns:**

I do not have concerns on the ethical implications of the work.

**Claims And Evidence:**

No

**Claims Explanation:**

The core idea of the paper is clear and the writing is mostly clean, but the evidence does not fully support the claims as stated, for the reasons below.

Pros

- Having a comparison of vanilla AlphaZero, multi-frame variant, and AZAL variant separates the representation bottleneck from a signal bottleneck.
- Choice of games, Connect Four and Chomp, is well motivated: one partisan game and one impartial game.
- The Grundy oracle gives Chomp an unambiguous optimality criterion, which is cleaner than most game-playing evaluations.
- The framing of the paper that strong play is not the same as perfect play, and that oracle-evaluable games make the gap measurable makes the paper easy to follow.

Cons

- The experimental results are too thin to carry the claims as stated, and this is my main concern. Results displayed in Table 1 comes from single deterministic greedy rollout but general claims are made (e.g., “AZAL reaches perfect play”). I don’t find these claims firmly supported by Table 1 results on its own.
- On Connect Four, AZAL has a higher first-player match rate than vanilla (0.905 vs 0.800) but a shorter longest chain (9 vs 14), which contradicts the text's claim that AZAL preserves the moveset longer. The longest-chain metric is also defined ambiguously.
- Go-Exploit is cited as motivation but not compared, even though it targets exactly the deep-state undersupervision the paper invokes. Other literature in AlphaZero with auxiliary supervision is not discussed at all.
- The discussion of the moving-target problem in Section 7 is plausible but entirely verbal. Claims like "losing positions during self-play often appear winning when the opponent fails to exploit them" can be measured directly from the logged traces (i.e., how often does self-play resolve a position against its oracle value?). Additional results should be reported.
- Reproducibility is limited: no code is provided, and key hyperparameters (MCTS simulations, network size, $\lambda_{aux}$, self-play volume, training length) are not reported.

**Requested Changes:**

As discussed above, the manuscript can benefit from fixing the “Cons” mentioned above.

---

> ### Author Response · Authors · 2026-07-01
> **Revisions for Reviewer 1oxy**
>
> Dear Reviewer,
>
> Thank you for your careful reading of our manuscript and for the constructive feedback. We appreciate your positive comments regarding the framing of strong versus perfect play, the choice of Connect Four and Chomp as oracle-evaluable domains, and the use of the Chomp Grundy oracle as an evaluator. We have revised the manuscript to address the main concerns raised, especially the lack of experimental evidence, the ambiguity around the longest-chain metric, the Connect Four interpretation, missing related-work discussion, the moving-target diagnostic, and reproducibility details.
>
> Reviewer Comment 1:
> The experimental results are too thin to carry the claims as stated, since Table 1 originally came from a single deterministic greedy rollout.
>
> Author response 1:
> Thank you for pointing this out. We agree the original deterministic trace evaluation was insufficient to support the broader claims stated. In the revised manuscript, we expanded the experimental evidence.
>
> We report multi-seed full-game oracle consistency over 60 self-play traces across three seeds for each model/game setting. We also added a random-start sampled-state evaluation to test if the observed behavior persists beyond a single opening rollout. The deterministic greedy traces are now presented as representative examples, not as the sole basis for the main claims.
>
> The revised manuscript reports that AZAL reaches perfect full-game oracle consistency on Chomp 10×11, high but incomplete consistency on Chomp 9×10, and improved but imperfect consistency on Connect Four.  The claims throughout the paper have been softened to avoid implying universal perfect play.
>
> The revised text in the Abstract states:
> “AZAL substantially improves oracle consistency across multi-seeded full-game traces and sampled-state evaluations. On Chomp, AZAL reaches perfect full-game oracle consistency on 10×11 and high but not complete consistency on 9×10; on Connect Four, AZAL improves oracle-match rate and delays the first oracle mistake, but does not reach perfect play.”
>
> Reviewer Comment 2:
> On Connect Four, AZAL has higher match rate than vanilla but a shorter longest chain, which contradicts the claim that AZAL preserves the moveset longer. The longest-chain metric is also ambiguous.
>
> Author response 2:
> We agree with this concern. In the revised manuscript, we clarified the definition of the longest oracle-consistent chain and corrected the interpretation of the Connect Four results.
> The longest-chain metric is defined as the longest contiguous run of oracle-consistent moves within the rollout. We also added FirstFail, the first labeled ply at which the agent makes a non-oracle move. This better captures if the agent preserves the early optimal line.
>
> For Connect Four, we no longer claim that AZAL improves the longest-chain metric. Instead, we state that AZAL improves oracle-match rate and delays the first oracle mistake, while the mean longest chain remains unchanged.
>
> The revised text now states:
> “From Tables 1 and 2, AZAL improves full-game match from 0.785 to 0.849, increases sampled-state match from 0.589 to 0.768, and delays the first failure from 0.867 to 5.283. However, AZAL does not improve the mean longest chain (16.967 vs. Vanilla 17.000) and has zero perfect traces. Thus, AZAL improves the density and timing of oracle-consistent decisions in Connect Four, but does not produce fully oracle-consistent play.”
>
> Reviewer Comment 3:
> Go-Exploit is cited as motivation but not compared, and other AlphaZero auxiliary-supervision literature is not discussed.
>
> Author response 3:
> Thank you for raising this concern. We revised the related-work discussion to clarify the relationship between our work and Go-Exploit. We agree that Go-Exploit targets an important and closely related issue: deep-state undersupervision. However, our current study is diagnostic rather than a sample-efficiency comparison. Our goal is to evaluate move-by-move oracle consistency under a shared AlphaZero self-play pipeline, not to benchmark against alternative training schedules.
>
> We explicitly acknowledge that we do not compare against Go-Exploit and that such a comparison would be valuable future work. We softened claims around AZAL to avoid implying that it is the definitive solution to deep-state undersupervision.
>
> The revised related work text states:
> “Go-Exploit targets deep-state undersupervision and sample efficiency. However, our work differs as it diagnoses oracle consistency move-by-move instead of proposing a sample-efficient training schedule. Thus, the goal is not to beat Go-Exploit but to test if standard self-play recovers exact oracle trajectories.”
>
> We also added the limitation in the Discussion:
> “We do not compare against Go-Exploit or other deep-state resampling methods; such comparisons would help separate data-distribution fixes from auxiliary-supervision fixes.”

---

> ### Author Response · Authors · 2026-07-01
> **Revisions for Reviewer 1oxy, Part 2**
>
> Reviewer Comment 4:
> The moving-target discussion in Section 7 is plausible but verbal. Claims such as losing positions appearing winning during self-play should be measured directly.
> Author response 4:
> We agree that this claim was under-supported in the original version. In response, we added a direct diagnostic measuring these cases, where the oracle states that the current player is losing, but the same player later wins the completed self-play rollout because the opponent fails to exploit the advantage.
>
> This is reported in the revised Discussion as Table 3, “Oracle value versus realized self-play outcome.” The results demonstrate that such target corruption does occur in our rollouts. For example, in Connect Four, losing-but-won positions occur for both Vanilla and AZAL. In Chomp 9×10, they also occur for AZAL. We revised the discussion to be more cautious: the diagnostic supports the existence of self-play target corruption, but it does not imply that AZAL always reduces these cases.
>
> The revised text states:
> “Table 3 shows this failure mode occurs in our rollouts: some oracle-losing positions are later won by the same player because the opponent fails to exploit the advantage. In Connect Four, this occurs for both Vanilla and AZAL: 0.144 and 0.174. In Chomp 9×10, it also appears 14% of the time for AZAL. These results support the existence of self-play target corruption, but not a simple monotonic story where AZAL always reduces losing-but-won cases.”
>
> Reviewer Comment 5:
> Reproducibility is limited: no code is provided, and key hyperparameters are not reported.
>
> Author response 4:
> Thank you for pointing this out. We have addressed this in two ways.
>
> First, we added an anonymous code link to the manuscript.
>
> Second, we added a reproducibility table in the appendix reporting the key hyperparameters for the main experiments.
>
> The revised manuscript now includes Table 5, “Reproducibility details for the main experiments.” We also state that all models use the final checkpoint and three training seeds, with evaluation seeds reported explicitly.
>
> Thank you again for the detailed feedback.
>
> Sincerely,
> The authors

---

> ### Comment · Reviewer_1oxy · 2026-07-11
> **Response to Author Rebuttal**
>
> Thank you for the thorough revision: the improved experiments, rescoped claims, and the added reproducibility details has addressed all of my main concerns.

---

### Review · Reviewer_4qEa · 2026-06-14

**Summary Of Contributions:**

The paper contributes a diagnostic study of the gap between strong play and perfect play in AlphaZero-style agents. Using exact oracles for Connect Four and Chomp, the authors show that vanilla AlphaZero can learn strong policies but still fail to follow oracle-optimal trajectories.

A key strength is the use of trace-level oracle evaluation, which makes these failures concrete. The main finding is that multi-frame inputs do not solve the issue, while the proposed AZAL auxiliary loss greatly improves oracle consistency; While a key weakness is that evaluation is mainly based on deterministic rollouts from fixed checkpoints, so the results are not yet a broad statistical study across seeds or states.

**Audience:**

Yes

**Audience Explanation:**

The findings may interest TMLR readers working on reinforcement learning, search, and game-solving. The paper highlights a useful gap between strong play and perfect play in AlphaZero-style agents, especially under sparse rewards and oracle-based evaluation.

**Broader Impact Concerns:**

No major concerns.

**Claims And Evidence:**

No

**Claims Explanation:**

The evidence is only partially convincing. The use of exact oracles makes the reported trace-level failures inspectable, but the empirical support is too limited to justify the broader claims, concretely:

- **Limited statistical support (Major):** The main trace-level results are based on deterministic greedy rollouts from fixed checkpoints, without multiple seeds, confidence intervals, or evaluation over a broader distribution of states. This makes it difficult to judge how robust or typical the reported failures and improvements are.
- **Limited external validity (Major):** The experiments cover only two small oracle-evaluable games and a small number of board settings. This supports a diagnostic case study, but not a broad conclusion about AlphaZero-style learning in sparsely rewarded games in general.
- **Claim strength vs. evidence (Minor):** The conclusion that the main bottleneck is the weakness of the standard AlphaZero search-learning signal is plausible, but not fully isolated experimentally. Other factors such as search budget, architecture, training stability, checkpoint selection, and hyperparameter sensitivity are not systematically ruled out. Can add some ablation studies on these.

Overall, the evidence provided in the paper is better viewed as supporting a focused case study rather than fully establishing the paper’s broader claims.

**Requested Changes:**

I suggest the following changes to improve the paper:

1. **Add multi-seed and checkpoint evaluation (critical).** For Connect Four, Chomp 9×10, and Chomp 10×11, report oracle-match rate and longest oracle-consistent chain over at least 3–5 random seeds and multiple checkpoints, with mean and standard deviation. This would show whether the reported vanilla failures and AZAL gains are stable rather than trajectory-specific.
2. **Evaluate more than one greedy rollout (critical).** In addition to the single rollout from the standard initial state, evaluate oracle consistency on a set of sampled states, e.g., Connect Four positions at different depths and Chomp states obtained after several random legal moves. This would make the trace-level claims less dependent on one deterministic trajectory.
3. **Add targeted sensitivity studies (important).** Run small sweeps over MCTS simulations, auxiliary-loss weight, and model size. For example, compare vanilla and AZAL under lower/higher search budgets and several values of λ_aux to test whether AZAL’s advantage is robust.
4. **Test on additional oracle-evaluable games or tasks (important).** Add at least one more sparse-reward or combinatorial game with an exact oracle, such as Nim or other impartial game with known optimal structure. This would help determine whether the observed failure mode and AZAL improvement are specific to Chomp/Connect Four or generalize to related settings.

---

> ### Author Response · Authors · 2026-07-01
> **Revisions for Reviewer 4qEa, Part 1**
>
> Dear Reviewer,
>
> Thank you for the careful and constructive review. We agree that the original version framed the results too broadly relative to the amount of evaluation presented. In response, we revised the manuscript to frame the work as a diagnostic study, added multi-seed aggregate evaluations, included random-start sampled-state evaluations, and softened claims about generality and causality. The revised paper now emphasizes that the evidence supports a focused case study in two oracle-evaluable games.
>
> Below is a point-by-point response to the requested changes:
>
> Reviewer Comment 1:
> Add multi-seed and checkpoint evaluation (critical). For Connect Four, Chomp 9×10, and Chomp 10×11, report oracle-match rate and longest oracle-consistent chain over at least 3–5 random seeds and multiple checkpoints, with mean and standard deviation.
>
> Author response 1:
> Thank you for identifying this as a central issue. We agree the original deterministic traces were insufficient to establish robustness.  In the revised manuscript, we added multi-seed full-game oracle evaluations for Connect Four, Chomp 9×10, and Chomp 10×11. The main full-game results aggregate 60 full-game self-play traces across three seeds per model and game setting.  Table 1 reports oracle-match rate, longest oracle-consistent chain, first failure, and perfect-trace rate. The revised protocol clarifies that the deterministic greedy traces are retained as representative examples; the main evidence comes from the aggregate multi-seed evaluations.
>
> The revised results reveal that the qualitative conclusions are stable across seeds: vanilla AlphaZero has zero perfect traces in all three main settings, Multi-Frame does not improve Chomp oracle consistency, and AZAL substantially boosts oracle consistency. For example, on Chomp 10×11, AZAL reaches 60/60 perfect traces and a 1.000 oracle-match rate, while Vanilla and Multi-Frame remain at 0/60 perfect traces. On Connect Four, AZAL improves match rate from 0.785 to 0.849 and delays the first oracle mistake from 0.867 to 5.283, but does not reach perfect play.
>
> We did not complete a full multiple-checkpoint evaluation in this revision. To avoid overstating the evidence, we explicitly state in the experimental protocol that only final checkpoints are evaluated and that checkpoint sensitivity is left for future work.  We also added this limitation to the Discussion.
>
> Reviewer Comment 2:
> Evaluate more than one greedy rollout (critical). In addition to the single rollout from the standard initial state, evaluate oracle consistency on a set of sampled states.
>
> Author response 2:
> We agree and have added this evaluation. The revised paper now includes random-start sampled-state oracle evaluations in Table 2. These states are generated by applying random legal moves and are sampled across three seeds. We evaluate only states for which the oracle-best moveset is defined, so the sampled-state metric directly measures the robustness of the learned policy.
>
> The sampled-state results support the same pattern as the full-game traces.  On Chomp 9×10, Vanilla reaches 0.500 sampled-state match, Multi-Frame reaches 0.176, and AZAL reaches 1.000. On Chomp 10×11, Vanilla reaches 0.400, Multi-Frame reaches 0.286, and AZAL reaches 0.829. On Connect Four, AZAL improves sampled-state match from 0.589 to 0.768.
>
> We also revised the wording around Figure 3 and the trace tables to clarify that these are representative oracle-annotated examples.
>
> Reviewer Comment 3:
> Add targeted sensitivity studies (important). Run small sweeps over MCTS simulations, auxiliary-loss weight, and model size.
>
> Author response 3:
> We agree that targeted sensitivity studies would strengthen the paper, especially for isolating if AZAL’s advantage persists across search budget, auxiliary-loss weight, and architecture scale.
> Unfortunately, we were not able to complete a full set of sensitivity sweeps in this revision within the review period. Instead, we made two changes to avoid overstating the result.
>
> First, we added a reproducibility table reporting the main architecture, MCTS budget, training iterations, replay buffer, batch size, learning rate, auxiliary-loss weight, temperature schedule, root noise, checkpoint rule, and seeds. This makes the experimental setting transparent and easy to reproduce.
>
> Second, we softened the causal interpretation throughout the manuscript. The revised paper says that the standard AlphaZero search-learning signal is a plausible bottleneck, rather than claiming that it is fully isolated as the only bottleneck. In the Discussion, we explicitly state that the results “support, but do not fully isolate” the weak-search-learning-signal explanation, and that future work should vary search budget, architecture, and checkpoint selection.

---

> ### Author Response · Authors · 2026-07-01
> **Revisions for Reviewer 4qEa, Part 2**
>
> Reviewer Comment 4:
> Test on additional oracle-evaluable games or tasks (important). Add at least one more sparse-reward or combinatorial game with an exact oracle, such as Nim or another impartial game.
>
> Author response 4:
> Thank you for this suggestion.  We agree that adding another oracle-evaluable game would improve external validity. In this revision, we did not add a third game because it requires substantially more compute and training time than was feasible within the review period.
>
> Instead of adding a shallow third domain, we prioritized strengthening the statistical support for the two existing domains.
>
> To address this concern directly, we revised the framing of the paper. The manuscript presents the results as a diagnostic case study in two oracle-evaluable domains, not as a broad claim about all AlphaZero-style sparse-reward games. The Discussion explicitly states that results should not be generalized beyond Connect Four and Chomp, and broader claims require additional games and larger settings.
>
> We also added a more careful discussion of why the Multi-Frame result in Chomp should not be read as contradicting the Nim-specific motivation from prior work.  In Nim, short histories expose simple heap-size changes, whereas in Chomp, moves remove two-dimensional sets and induce nonlocal changes in the reachable state space. Therefore, our Multi-Frame conclusion is restricted to rectangular Chomp.
>
> Reviewer Comment 5:
> The conclusion that the main bottleneck is the weakness of the standard AlphaZero search-learning signal is plausible, but not fully isolated experimentally.
>
> Author response 5:
> We agree. In the revised manuscript, we softened this claim substantially. The paper now argues that the weak search-learning signal is a plausible bottleneck, supported by the contrast between Vanilla, Multi-Frame, and AZAL under the same self-play + MCTS framework. We no longer present this as a fully isolated causal mechanism.
>
> The Discussion explicitly acknowledges that search budget, architecture, training stability, and checkpoint selection are not systematically ruled out.
>
> We revised the conclusion to state that AZAL “substantially improves” oracle consistency but is not a complete solution. The revised manuscript therefore distinguishes between strong improvement and complete recovery.
>
>
> Reviewer Comment 6:
> The experiments cover only two small oracle-evaluable games and a small number of board settings. This supports a diagnostic case study, but not a broad conclusion about AlphaZero-style learning in sparsely rewarded games in general.
>
> Author response 6:
> We agree with this assessment and revised the manuscript accordingly. The paper frames the contribution as a controlled diagnostic study, not as a broad generalization claim. In the Introduction, we emphasize that oracle-evaluable games provide a useful diagnostic setting for separating strong play from exact optimality. In the Discussion, we explicitly state the results are limited to two small oracle-evaluable games and should be interpreted as a diagnostic case study.
>
>
> Thank you again for the helpful feedback.
>
> Sincerely,
> The authors

---

> > ### Comment · Reviewer_4qEa · 2026-07-13
> > **Response to author rebuttal**
> >
> > Thank you for the revision. The strengthened evaluation, more appropriately scoped claims, and added reproducibility details address my main concerns.

---

### Review · Reviewer_NwDS · 2026-06-17

**Summary Of Contributions:**

The paper asks if AlphaZero's strong play is the same as perfect play. It checks this move by move in two oracle-evaluable games. Connect Four is a solved partisan game, and every move has an exact game-theoretic value. Chomp is an impartial game, and optimal play means moving to a g=0 Grundy position. All three methods share one self-play + MCTS pipeline. The methods are vanilla AlphaZero, a multi-frame variant, and AZAL. AZAL adds an oracle-derived auxiliary policy loss. It does not change the search or the value targets.

The empirical story is simple. Vanilla AlphaZero plays strongly, but it does not stay on the optimal trajectory. In Connect Four it leaves the winning line. On the larger Chomp boards it stops restoring the g=0 invariant. The multi-frame input does not fix this. AZAL does. It matches the oracle on every checked move in the reported Chomp traces, and almost every move in the reported Connect Four trace. The representation change fails, but the stronger supervision works. From this the authors argue the real bottleneck is the weak learning signal in standard AlphaZero. It is not the network's ability to represent good play.

**Audience:**

Yes

**Audience Explanation:**

Whether superhuman AlphaZero-style agents actually play optimally is an active question at top venues. Wang et al. [1] showed that superhuman Go AIs have exploitable blind spots and can be tricked into serious blunders, the same strong-versus-optimal gap this paper studies. Hamrick et al. [2] found that in MuZero the benefit of planning is mainly in learning, and that planning alone does not drive strong generalization, which connects to this paper's claim that the bottleneck is the search-learning signal. So the findings here would interest readers in self-play RL and combinatorial games.


[1]: Adversarial Policies Beat Superhuman Go AIs, Wang et al., ICML 2023.

[2]: On the Role of Planning in Model-Based Deep Reinforcement Learning, Hamrick et al., ICLR 2021.

**Claims And Evidence:**

No

**Claims Explanation:**

The per-move oracle checks are exact, and I trust the labels in Tables 2, 3, and 4. So the existence claims hold: vanilla AlphaZero can play strongly yet still leave the optimal trajectory, and AZAL can match the oracle on the shown traces. These parts are clear and convincing.

The problem is the comparative and interpretive claims. Every reported number comes from one deterministic greedy rollout of one training seed. A seed does not just change one policy update. It sets the network init, the Dirichlet noise, and the sampled self-play games, so a new seed gives a different self-play distribution and a different final checkpoint. The defense that rollouts are deterministic only removes evaluation noise, not this training-run noise. The abstract and conclusion say AZAL "substantially improves" and that multi-frame fails. That reads stronger than one seed can show.

The rates also come from very few moves. The 9x10 vanilla Match1 of 0.600 is only 3 of 5 player-1 labeled moves, not a stable estimate. And AZAL's player-2 match rate of 0.000 on both Chomp boards is misleading: every player-2 move is from a g=0 state with no oracle best move, so it is a side effect of player 1 playing perfectly, not a real failure.

The main claim, that the bottleneck is the weak learning signal and not the representation, rests on one negative result (multi-frame) and one positive result (AZAL). The moving-target story is read off the loss oscillations, never shown directly.

**Requested Changes:**

Support the method comparison, or soften it. Run more training seeds and report the spread, or evaluate each trained network over many positions instead of one greedy rollout. Otherwise, tune down the comparative and interpretive claims in the abstract, Section 6, and Section 8, and frame the paper as an existence-plus-mechanism study. As written, the ranking of vanilla, multi-frame, and AZAL rests on a single seed.

Minor:

* Table 1: please report the number of oracle-labeled moves behind each rate. For example, the 9x10 vanilla Match1 of 0.600 is 3 of 5 player-1 moves (Table 2b), so showing the denominator makes the small sample explicit.

* Table 1 and Section 6.1: multi-frame is evaluated on Chomp only, with no Connect Four row. Please add a Connect Four multi-frame result, or scope the claim that multi-frame does not help to Chomp.

---

> ### Author Response · Authors · 2026-07-01
> **Revisions for Reviewer NwDS, Part 1**
>
> Dear Reviewer,
>
> Thank you for carefully reading our manuscript, “AlphaZero in Sparsely Rewarded Games: Limits and Auxiliary Supervision.” We appreciate the time and effort you dedicated to evaluating the paper and are grateful for the detailed comments.
>
> In the revised manuscript, we have addressed the main concern of the original version: relying heavily on a single deterministic rollout. The revised results now include multi-seed full-game evaluations, with 60 full-game traces across three training seeds for each evaluated model/game setting. We also included random-start sampled-state oracle evaluations to test robustness of the models.  The numerator/denominator counts for oracle-match rates are reported, and the interpretation of the weak-learning-signal explanation is softened.
>
> Below is a point-by-point response to the requested changes:
>
> Reviewer Comment 1:
> The per-move oracle checks are exact, and I trust the labels in Tables 2, 3, and 4. So the existence claims hold: vanilla AlphaZero can play strongly yet still leave the optimal trajectory, and AZAL can match the oracle on the shown traces. These parts are clear and convincing.
>
> Author response 1:
> Thank you for this assessment.  We have preserved this core framing in the revision: the paper studies the distinction between strong empirical play and exact oracle-consistent play in two oracle-evaluable games.
>
> Reviewer Comment 2:
> The problem is the comparative and interpretive claims. Every reported number comes from one deterministic greedy rollout of one training seed. A seed does not just change one policy update. It sets the network init, the Dirichlet noise, and the sampled self-play games, so a new seed gives a different self-play distribution and a different final checkpoint. The defense that rollouts are deterministic only removes evaluation noise, not this training-run noise. The abstract and conclusion say AZAL “substantially improves” and that multi-frame fails. That reads stronger than one seed can show.
>
> Author response 2:
> We agree with the reviewer. The original version relied heavily on one deterministic greedy rollout, which removed evaluation noise but did not address training-run variations.
>
> To address this, we revised the main empirical evaluation. The revised manuscript now reports multi-seed full-game oracle consistency. Specifically, the main results aggregate 60 full-game self-play traces across three training seeds for each evaluated model/game setting. We also added a separate random-start sampled-state oracle evaluation to test if the observed behavior persists beyond standard rollouts.
>
> In addition, we softened the comparative and interpretive language throughout the manuscript. We no longer claim that the weak AlphaZero learning signal is definitively the bottleneck. Instead, we frame it as a plausible bottleneck supported by the observed comparison between vanilla AlphaZero, multi-frame inputs, and AZAL.
>
> The revised manuscript states that AZAL improves oracle consistency across multi-seeded full-game traces and sampled-state evaluations, emphasizing that AZAL does not reach perfect play in every setting.
>
> Reviewer Comment 3:
> The rates also come from very few moves. The 9x10 vanilla Match1 of 0.600 is only 3 of 5 player-1 labeled moves, not a stable estimate.
>
> Author response 3:
> Thank you for pointing this out. We agree that the original table presentation made small-denominator rates appear more stable than they were.
>
> We revised the tables so oracle-match rates include the corresponding numerator and denominator counts. For example, the revised main results table reports Chomp 9×10 Vanilla as: Match = 0.609 (487/800)
>
> We also added a player-specific oracle-match table in the appendix, where Match1 and Match2 are reported with numerator/denominator counts. This makes the number of labeled decisions behind each rate explicit.

---

> ### Author Response · Authors · 2026-07-01
> **Revisions for Reviewer NwDS, Part 2**
>
> Reviewer Comment 4:
> AZAL’s player-2 match rate of 0.000 on both Chomp boards is misleading: every player-2 move is from a (g=0) state with no oracle best move, so it is a side effect of player 1 playing perfectly, not a real failure.
>
> Author response 4:
> We agree. The reviewer is correct that in Chomp, when a state has Grundy number (g=0), there is no oracle-best move under our winning-position oracle definition. Reporting these cases as failures was misleading.
>
> We revised the player-specific Chomp results accordingly. In the revised table, when no labeled oracle decision is defined, the entry is reported as N/A rather than as a failed match. For example, AZAL on Chomp 10×11 now reports Match2 as N/A, because player 2 only moves from (g=0) states under perfect player-1 play.
>
> The revised appendix explicitly states that Chomp states with (g=0) have no oracle-best move, so some player-specific entries are not defined.
>
>
> Reviewer Comment 5:
> The main claim, that the bottleneck is the weak learning signal and not the representation, rests on one negative result, multi-frame, and one positive result, AZAL. The moving-target story is read off the loss oscillations, never shown directly.
>
> Author response 5:
> We agree that the original wording was too strong. We have revised the manuscript to avoid claiming the weak-learning-signal explanation is fully isolated.
>
> The revised manuscript now describes the weak search-derived supervision as a plausible limiting factor, not a definitive causal conclusion. We also added a diagnostic table measuring cases where the oracle says the current player is losing, but the same player later wins the completed self-play rollout. This provides more direct evidence that self-play can generate misleading learning targets when the opponent fails to exploit an oracle-winning advantage.
> At the same time, we explicitly acknowledge that this diagnostic does not fully isolate the mechanism. The revised discussion now states that the results support the existence of self-play target corruption, but not a simple monotonic story in which AZAL always reduces such cases.
>
> We also added a limitation noting that future work should vary search budget, architecture, and checkpoint selection to more fully isolate the mechanism.
>
> Requested Change 1:
> Support the method comparison, or soften it. Run more training seeds and report the spread, or evaluate each trained network over many positions instead of one greedy rollout. Otherwise, tune down the comparative and interpretive claims in the abstract, Section 6, and Section 8, and frame the paper as an existence-plus-mechanism study. As written, the ranking of vanilla, multi-frame, and AZAL rests on a single seed.
>
> Author response to change 1:
> We have addressed this requested change in two ways.
>
> First, we strengthened empirical support. The revised manuscript reports results across three training seeds and 60 full-game traces for each evaluated model/game setting. We added sampled-state oracle evaluation from random-start positions, so the conclusions do not depend only on one standard-start greedy rollout.
>
> Second, we softened the interpretation. The abstract, Section 6, Discussion, and Conclusion now frame the paper as a diagnostic study of strong play versus perfect play. We describe AZAL as improving oracle consistency in the tested settings, while emphasizing it is not a complete solution and the weak-learning-signal explanation remains plausible.
>
> Requested Change 2:
> Table 1: please report the number of oracle-labeled moves behind each rate. For example, the 9x10 vanilla Match1 of 0.600 is 3 of 5 player-1 moves, so showing the denominator makes the small sample explicit.
>
> Author response to change 2:
> We have revised the tables to include numerator/denominator counts behind the reported rates.
> The revised main table now reports pooled oracle-match counts, such as: Match = 0.609 (487/800)
>
> The appendix table also reports player-specific rates with counts. This makes the denominator behind each rate explicit and avoids overstating small-sample estimates.
>
> Requested Change 3:
> Table 1 and Section 6.1: multi-frame is evaluated on Chomp only, with no Connect Four row. Please add a Connect Four multi-frame result, or scope the claim that multi-frame does not help to Chomp.
>
> Author response to change 3: We agree. Since multi-frame AlphaZero was evaluated only on Chomp, the original wording should not have implied a broader cross-domain conclusion.
>
> We revised the manuscript to explicitly scope the multi-frame claim to Chomp. The abstract now identifies the multi-frame variant as limited to Chomp, and Section 6.1 now states:
>
> “We evaluate multi-frame only on Chomp, so our multi-frame conclusion is restricted to the rectangular Chomp settings.”
>
> Thus, the revised manuscript no longer claims that multi-frame fails in Connect Four.
>
>
> Thank you for your time and effort.
>
> Sincerely,
> The authors

---

> > ### Comment · Reviewer_NwDS · 2026-07-12
> >
> > Thank you for the careful and substantive revision. The strengthened evaluation, the more precisely scoped claims, and the new protocol details together resolve the main issues I raised.